# The planarian wound epidermis gene *equinox* is required for blastema formation in regeneration

M. Lucila Scimone[1,2,5], Jennifer K. Cloutier [2,3,4,5], Chloe L. Maybrun[2,3] & Peter W. Reddien [1,2,3✉]

Regeneration often involves the formation of a blastema, an outgrowth or regenerative bud formed at the plane of injury where missing tissues are produced. The mechanisms that trigger blastema formation are therefore fundamental for regeneration. Here, we identify a gene, which we named *equinox*, that is expressed within hours of injury in the planarian wound epidermis. *equinox* encodes a predicted secreted protein that is conserved in many animal phyla. Following *equinox* inhibition, amputated planarians fail to maintain wound-induced gene expression and to subsequently undergo blastema outgrowth. Associated with these defects is an inability to reestablish lost positional information needed for missing tissue specification. Our findings link the planarian wound epidermis, through *equinox*, to regeneration of positional information and blastema formation, indicating a broad regulatory role of the wound epidermis in diverse regenerative contexts.

---

[1] Howard Hughes Medical Institute, Massachusetts Institute of Technology, Cambridge, MA 02139, USA. [2] Whitehead Institute for Biomedical Research, Cambridge, MA 02142, USA. [3] Department of Biology, Massachusetts Institute of Technology, Cambridge, MA 02139, USA. [4] Harvard/MIT MD-PhD, Harvard Medical School, Boston, MA 02115, USA. [5] These authors contributed equally: M. Lucila Scimone, Jennifer K. Cloutier. ✉email: reddien@wi.mit.edu

Most animals face the challenge of repairing tissues following injury. Whereas some animals can only heal minor wounds, others can regenerate entire appendages or major parts of body axes[1,2]. In many highly regenerative animals, regeneration involves cell proliferation near the wound and formation of a blastema—an outgrowth that forms at the wound site and where differentiation of many missing tissues takes place. How regeneration initiates through blastema formation is not fully understood. Regeneration in some vertebrates requires the formation of a wound epidermis (WE), a specialized epidermis that covers the amputation site soon after injury[3–7]. Following amputation, intact epithelial cells detach from the basement membrane and migrate to cover the wound surface[8]. Once the wound is covered, epithelial cells from the early wound epidermis proliferate to form a thick stratified epithelium, mobilization of progenitors occurs (involving tissue-resident stem cells and/or dedifferentiation), and the regeneration blastema forms[8,9]. This thick epidermis functions as a signaling center to allow blastema growth[10,11]. Disruption or surgical removal of the WE in salamanders results in regeneration failure[5,12].

Planarian regeneration also involves the formation of a blastema, as well as changes in pre-existing tissues. Planarians are able to regenerate from a large array of injuries, including from tiny fragments of the body. This involves pluripotent stem cells called neoblasts, which reside in a mesenchymal compartment and generate all new planarian tissues[13]. Following amputation, the wound surface in planarians is covered by spreading and migrating epidermal cells[14–16]. This is followed by the formation of a blastema that is produced from proliferating neoblasts near the wound. However, whether wound epithelialization has a regulatory role in planarian regeneration by promoting blastema formation is unknown.

Here, we identify a gene, which we named *equinox*, that encodes a predicted secreted protein conserved in many metazoan clades. *equinox* is expressed in the planarian wound epidermis shortly after injury. Following *equinox* inhibition, animals were unable to regenerate. Inhibition of *equinox* affected maintenance of wound signaling and the associated resetting of positional information needed to specify the identity of missing tissues. Neoblasts also failed to display upregulation of proliferation at wounds. Importantly, the regeneration failure observed following *equinox* inhibition only occurred in injury contexts requiring blastema formation, where positional information regeneration and blastema outgrowth bring about regeneration. This work defines a role for the wound epidermis and *equinox* in initiating regeneration through blastema formation.

## Results and discussion

**bmp4, a DV-patterning factor, is required for regeneration.** Planarian RNA interference (RNAi) studies have identified many genes required for regeneration, but most of these genes are also required for tissue turnover in uninjured animals, such as genes essential for neoblast biology. Neoblasts are constantly dividing to enable turnover of all tissues as part of natural homeostasis[13]; consequently, perturbation of neoblast biology leads to lack of animal viability. Few genes are known to be required for regeneration but not cell turnover, and such genes are good candidates to mediate regeneration-specific mechanisms. Bmp signaling has a prominent role in controlling patterning of the dorsal–ventral (DV) axis in many organisms, including planarians[17–19] (Supplementary Fig. 1a). Prior work showed that inhibition of Bmp signaling in planarians (e.g., with *bmp4* or *smad1* RNAi) blocks regeneration of the medial–lateral (ML) axis (after sagittal or parasagittal amputation) without inhibiting tissue turnover[17,18], however the reason for this defect is unknown. Using RNAi methods improved since these original studies (see the "Methods"

section), we now found that *bmp4* RNAi animals also failed to form head or tail blastemas after transverse amputation even after prolonged periods of time (20 days post amputation, dpa, Fig. 1a). This indicated the existence of a general requirement for *Bmp* signaling in planarian blastema formation.

The planarian body plan includes specialized cells and patterns of gene expression at the dorsal–ventral median plane at the animal margin (the lateral edge), sometimes referred to as the DV boundary (DVB). DVB-specific cells include specialized *laminB+* epidermal cells, which is a proxy for an intact DVB. *laminB+* epidermal cells did not regenerate in *bmp4* RNAi animals following transverse amputations (Fig. 1a and Supplementary Fig. 1b, c), similar to a defect previously observed for lateral regeneration following parasagittal amputation after Bmp-pathway inhibition[17,18]. Clusters of specialized muscle cells, with organizer-like activity influencing blastema pattern, found at the head and tail tips, are called anterior and posterior poles, respectively. Poles are specified from neoblasts and pole progenitors coalesce during regeneration at the DVB and at the pre-existing midline of the amputated fragment[20–23]. *bmp4* RNAi tail fragments frequently failed to regenerate an anterior pole (Supplementary Fig. 1c). However, this regeneration defect was incompletely penetrant, with an anterior pole forming in one third of the tail fragments (Supplementary Fig. 1c, d). Anterior poles formed more often at anterior-facing wounds of *bmp4* RNAi trunk fragments than at anterior-facing wounds of tail fragments (Supplementary Fig. 1d), but these wounds still lacked blastema outgrowth. Posterior poles formed frequently, even though no blastema was observed at *bmp4* RNAi posterior-facing wounds (Fig. 1a and Supplementary Fig. 1b, d). In cases where a pole formed in *bmp4* RNAi animals, it formed asymmetrically—instead of forming at the pre-existing midline, poles were shifted laterally and juxtaposed to the pre-existing lateral DVB (Fig. 1a and Supplementary Fig. 1b–d). The DVB has been associated with pole progenitor coalescence in prior work[20]. Small patches of new *laminB+* cells occasionally formed when *bmp4* RNAi animals were able to partially regenerate (Supplementary Fig. 1b). In *bmp4* RNAi animals that did form an anterior pole (a minority of tail fragments, or most trunk fragments), brain tissue, eyes, and pharynges were sometimes formed despite animals lacking blastemas (Supplementary Fig. 1b, c). This was similar to a phenotype previously observed following inhibition of *smad4*, a gene encoding a co-Smad associated with multiple classes of Tgf-β signaling[17]. These results raise the possibility that Bmp signaling is required for most-to-all contexts of blastema formation, rather than just ML regeneration.

**bmp4 is required for regeneration initiation.** Regeneration in planarians involves several phases. Following wound closure by epidermal cells, there is an initial "wound response" (0.5–12 h post amputation, hpa) that occurs at essentially all injuries and involves wound-induced gene expression, an increase in neoblast proliferation, and elevated apoptosis near the wound[24–27]. At injuries that remove substantial tissue, additional events subsequently occur that collectively comprise the "missing tissue response" or MTR (~16–48 hpa). These events include persistent wound-induced gene expression, a second and sustained phase of neoblast proliferation and accumulation at wounds, and body-wide elevation in levels of apoptosis[24,26–30]. Concurrent with the MTR, adult planarian positional information—in the form of position control gene (PCG) expression domains—regenerates in amputated fragments[31–33]. PCGs are regionally expressed genes along body axes that are associated with planarian patterning. PCGs are mostly expressed in muscle cells[31,34]. Following amputation, missing PCG expression domains are activated near

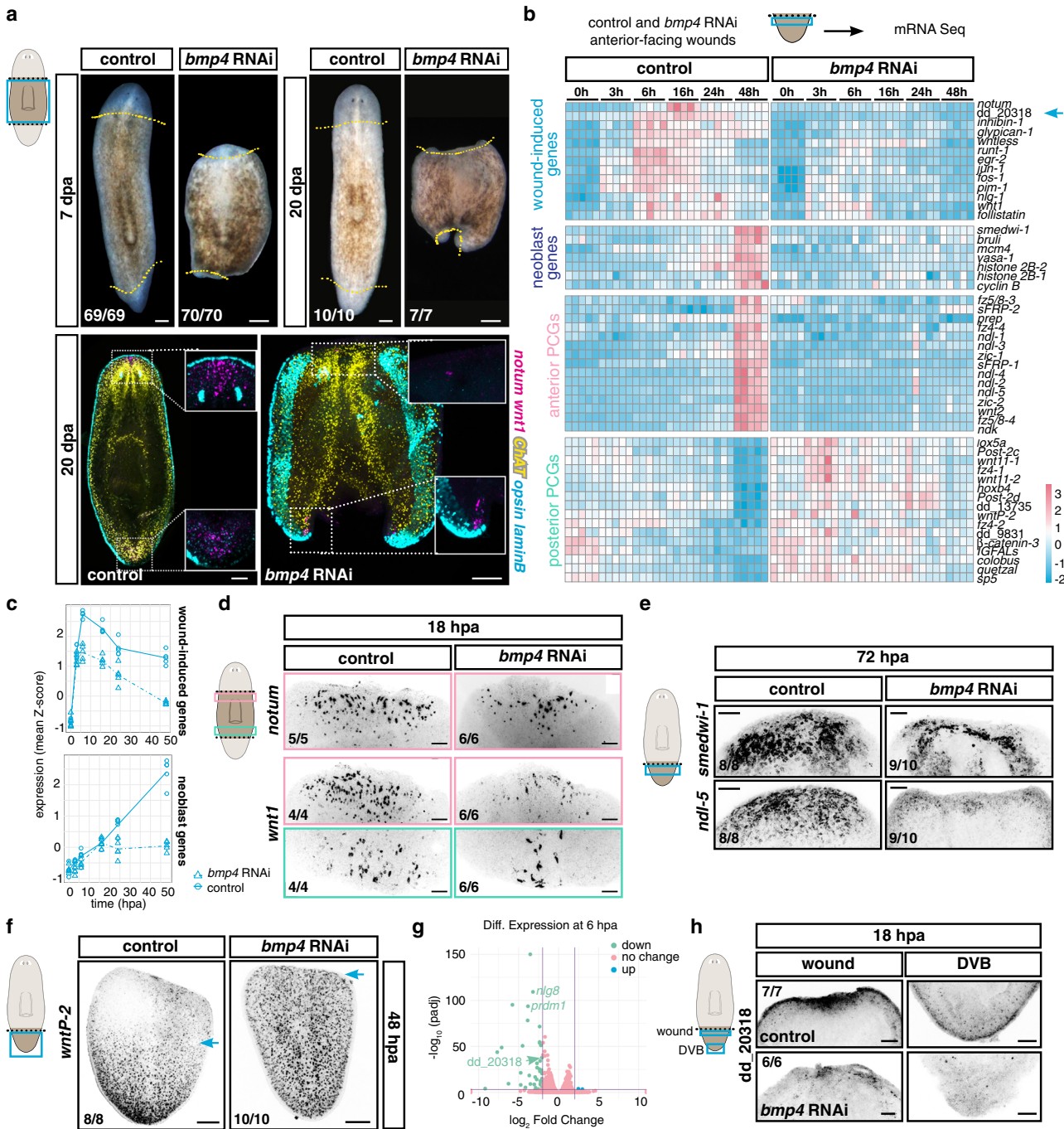

**Fig. 1 bmp4 is required for regeneration initiation. a** *bmp4* RNAi animals do not regenerate. *n* = 6 (left) and *n* = 2 (right and bottom) independent experiments. Dotted yellow line, amputation plane. **b** Heatmap shows expression of wound-induced genes, neoblast markers, anterior and posterior PCGs after *bmp4* RNAi. Some posterior-PCG labeled genes have not been connected to a signaling pathway but are regionally expressed in muscle. **c** Graphs show decreased expression of wound-induced genes and neoblast-specific genes in *bmp4* RNAi animals. **d** Reduced wound-induced gene expression, **e** reduced neoblast accumulation (*smedwi-1*) and expression of anterior markers (*ndl-5*) after *bmp4* RNAi. **f** Failed posterior PCG rescaling (*wntP-2*) in *bmp4* RNAi animals. Blue arrow, anterior expression edge of *wntP-2*. **g** Volcano plot shows dd_20318 expression is significantly lower in *bmp4* RNAi animals than in controls. **h** Reduced dd_20318 expression at 18 hpa in *bmp4* RNAi animals. Results shown in **d–f**, **h** are from two independent experiments. Colored boxes, area depicted in pictures. Scale bars, 100 μm.

wounds and some remaining PCG expression domains in an amputated fragment recede from the wound[31–33]. After the initiation of these changes, new differentiated cell types emerge in the blastema (~36–72 hpa) during a growth phase. PCG expression changes are required for regeneration, blastema patterning, and for the re-scaling of preexisting tissues to accommodate the smaller amputated fragment size[13].

We reasoned that some aspects of the early steps of regeneration might require *bmp4*. To assess this possibility, we performed bulk RNA sequencing of control and *bmp4* RNAi anterior-facing wounds from tail fragments at different time points following amputation (Supplementary Data 1). *bmp4* RNAi animals activated the early wound-induced gene expression response (3-6 hpa) but were unable to maintain such expression at later time points

(Fig. 1b, c and Supplementary Fig. 2a). Fluorescence in situ hybridization (FISH) experiments in trunk fragments showed similar results with an independent method (Fig. 1d and Supplementary Fig. 2b). We also assessed levels of canonical neoblast transcripts, including for the *piwi* homolog gene *smedwi-1*, which increase at wounds because of elevated neoblast proliferation and accumulation as part of the MTR. No increase was observed at 48 hpa in *bmp4* RNAi animals, suggesting a lack of neoblast accumulation (Fig. 1b, c), and FISH experiments supported this conclusion (Fig. 1e). Finally, *bmp4* RNAi animals failed to regenerate positional information during head regeneration—tail fragments failed to activate anterior PCG expression and failed to downregulate posterior PCGs at wounds (Fig. 1b, e, f and Supplementary Fig. 2c). The formation of posterior poles, albeit with abnormal location, at posterior-facing wounds indicates posterior positional information regeneration was less affected than anterior positional information regeneration; however, blastema outgrowth was still inhibited at both anterior-facing and posterior-facing wounds. These data indicate that *bmp4* RNAi animals failed to reset positional information needed for anterior regeneration and lacked a clear MTR. Taken together, our results show a requirement for Bmp signaling in regeneration initiation and blastema formation.

Genes required specifically for regeneration—but not for the capacity to generate cells during tissue turnover—are rare, and include *myoD* and *follistatin*[35–38]. Regeneration-specific defects caused by inhibition of *myoD* and *follistatin* exhibit some similarity to the *bmp4* RNAi phenotype. Inhibition of each gene causes lack of a regenerative response: defects in maintenance of wound-induced gene expression, perturbed neoblast accumulation, and failed positional information regeneration. However, differences between these phenotypes are also apparent and help in understanding the mechanistic basis for regeneration initiation. *myoD* inhibition affects activation of the muscle wound response specific to longitudinal fibers, including the expression of the genes *notum* and *follistatin*, but does not affect activation of the wound response in other muscle or epidermal cells[35]. This resulted in a failure to regenerate PCG expression domains, suggesting that resetting positional information is required for regeneration. *follistatin* is required for the MTR, but is not required for the initial activation of most wound-induced genes[36,38]. At many wound types, *follistatin* RNAi animals show only slow and delayed blastema formation, with missing tissues eventually fully returning in a blastema by 20 dpa[38]. By contrast, *follistatin* is required for head regeneration at many amputation planes (except at planes in the very anterior of the animal)[36–38]. This head-regeneration defect is explained by abnormal *wnt1* expression levels at wounds of *follistatin* RNAi animals: *follistatin* RNAi causes *wnt1* to be overexpressed at wounds and the high level of *wnt1* causes a failure in PCG re-setting at anterior-facing wounds, blocking regeneration. When *follistatin* and *wnt1* are simultaneously inhibited, head regeneration proceeds, but slowly and without elevation of neoblast proliferation at wounds[38]. These prior findings indicate that blocking the MTR (and the associated elevation in neoblast proliferation) alone does not inhibit regeneration. These results indicate that PCG expression domain regeneration, but not elevation of neoblast proliferation at wounds, is required for regeneration. *bmp4* RNAi animals did not form blastemas even by 20 dpa (Fig. 1a), and failure in blastema growth occurred at different sites across the AP axis (Supplementary Fig. 3a). We suggest that defects in PCG regeneration in *bmp4* RNAi animals partially explain the requirement for *bmp4* in regeneration; however, blastema formation also failed in the cases where some PCG re-setting still occurred and new tissues formed in the context of preexisting tissue in these animals. Furthermore, why would a DV axis-patterning pathway be required for regeneration of PCGs on the AP axis? We considered the possibility that the

regeneration-specific defect in *bmp4* RNAi animals might involve a previously uncharacterized process associated with regeneration initiation.

**bmp4 expression is required for activation of equinox.** Whereas *bmp4* was required for the maintenance of wound-induced expression for many genes, the gene dd_20318 stood out as being dependent on *bmp4* for its initial activation at wounds, rather than just for maintenance of its expression (Fig. 1b). Furthermore, although dd_20318 showed significantly lower expression in *bmp4* RNAi animals at 6 hpa, it was unaffected in *myoD* or *follistatin* RNAi animals at a similar timepoint (Fig. 1g and Supplementary Fig. 3b). This raised the possibility that activation of dd_20318 at wounds is a component of the unique requirement for *bmp4* in regeneration. FISH experiments confirmed that dd_20318 activation at wounds was reduced in *bmp4* RNAi animals at 6 and 18 hpa (Fig. 1h and Supplementary Fig. 3c, d).

dd_20318 encodes a predicted secreted protein with two thrombospondin 1 domains (TSP1) and several EGF and EGF-like domains in the C-terminal portion of the protein (Fig. 2a). TSP1 domains were first identified in thrombospondins, a conserved family of extracellular glycoproteins involved in wound healing, angiogenesis, and connective tissue organization, all of which require cell–cell and cell–extracellular matrix interactions[39]. TSP1 domains are also found in non-thrombospondin proteins. Thrombospondins, which appear to have been lost in planarians and nematodes, include other domains (e.g., TSP3, TSPC) not present in the predicted dd_20318 protein. Proteins with similarity to dd_20318 protein sequence and domain structure are found in several phyla spanning much of the Bilateria (from cnidarians to cephalochordates), but are absent in vertebrates (Fig. 2a and Supplementary Fig. 4, Supplementary Data 2), suggesting that these proteins define a conserved family. A different protein called von Willebrand factor D and EGF-domains (VWDE), also shows similarity to the dd_20318 protein. This similarity was primarily observed in the EGF repeats, and VWDE proteins have differences in the remainder of their domain architecture with dd_20318 (Fig. 2a, Supplementary Fig. 4 and Supplementary Data 2). Specifically, VWDE proteins have a von Willebrand domain and lack TSP1 domains. Of note, *vwde* is expressed in regenerative blastemas in vertebrates and has a role in blastema growth in these animals[40]. It is possible that dd_20318 and VWDE protein classes have some functional similarities, but this will require further investigation to assess. We named the dd_20318 gene *equinox* for its expression and role in planarian regeneration (see below).

**equinox is expressed in the wound epidermis.** *equinox* was expressed in uninjured animals in the epidermis near the DVB (Fig. 2b). Cross sections showed that *equinox* was expressed dorsal to *laminB*+ epidermal DVB cells (Fig. 2b). We performed hematoxylin and eosin stainings as well as FISH in sagittal sections to further understand epidermal behavior and *equinox* expression during regeneration. Spreading of epidermal cells that partially covered the wound was observed by 6 hpa and became more apparent at 18 hpa (Fig. 2c), in agreement with previous TEM studies of planarian wound dynamics involving epidermis[14,15]. *equinox* expression covered the amputation plane by 6 hpa. At 18 hpa, a thicker layer of *equinox*+ cells was observed (Fig. 2c). To determine the cell-type specificity of *equinox* expression in regeneration, we first performed single-cell RNA sequencing (scRNA-seq) of anterior-facing wounds from tail fragments at 0, 6, and 16 hpa (Supplementary Fig. 5). *equinox* expression was enriched in epidermal cells but was also expressed in some *smedwi-1*+ cells (Fig. 3a). Although *smedwi-1* is a neoblast marker, *smedwi-1* transcripts can be residually detected in post-mitotic progenitors[41–43]. A

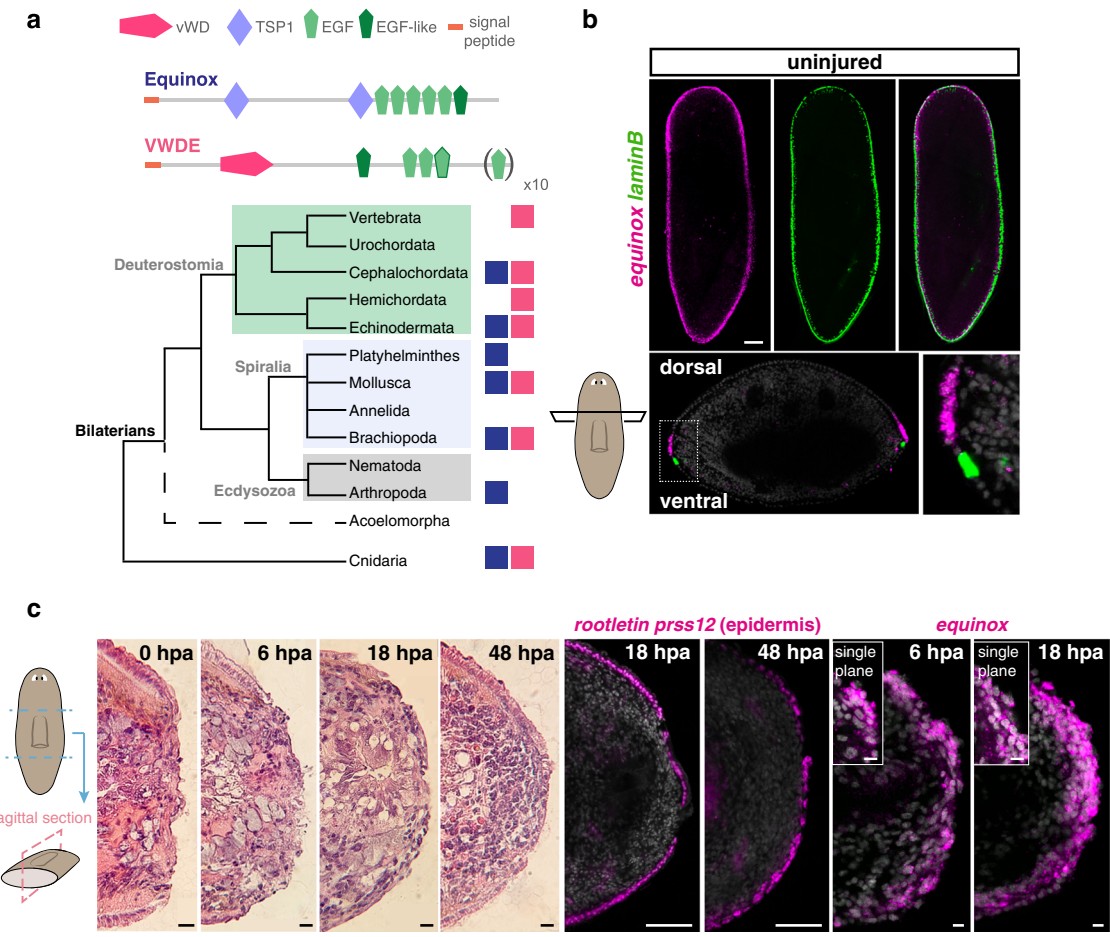

**Fig. 2 equinox is wound-induced in planarians. a** Domain structure and evolutionary conservation of Equinox and VWDE. Presence of an Equinox-like protein (blue squares) and VWDE (pink squares). **b** equinox is expressed dorsal to laminB+ DVB epidermal cells in uninjured animals. n = 3 independent experiments. **c** H&E staining of sagittal sections shows epidermal wound covering of transverse wounds (left, n = 1 experiment). Mature epidermis (prss12+, rootletin+) is observed by 18 hpa at the wound (middle). Wound-induced equinox expression following amputation is observed at the outer layer (right). n = 3 independent experiments.

small fraction of equinox-expressing cells was also observed in neurons (Fig. 3a). Because bmp4 is expressed in muscle cells and equinox is expressed in epidermis, no expression correlation between these two genes was observed (Supplementary Fig. 5).

Because equinox expression was observed in the epidermis and in smedwi-1+ cells in the scRNA-seq data, we asked whether these smedwi-1+ cells were epidermal progenitors. The planarian epidermal lineage is well characterized, with markers for different maturation stages known[44–47]. equinox was expressed throughout the entire epidermal lineage in regeneration, from zeta-neoblasts to early post-mitotic progenitors (prog-1+) to fully differentiated epidermal cells (prss12+, rootletin+) (Fig. 3b, Supplementary Fig. 6). Irradiation, which depletes neoblasts and their post-mitotic progeny cells, severely reduced equinox expression at 18 hpa, further indicating that equinox expression at this time point was enriched in epidermal progenitors (Fig. 3c). Expression of equinox at 6 hpa did not overtly change following irradiation, suggesting that expression at this early post-injury time point prominently included differentiated cells. Double FISH experiments using an equinox RNA probe together with probes for prog-1 (early-stage epidermal marker) or prss12 and rootletin (mature epidermal markers) and the SMEDWI-1 antibody to detect cells that are recently derived from neoblasts, showed cells co-expressing equinox and prog-1 at 6 hpa. Many equinox+ cells at this time, however, were prss12+ and rootletin+, consistent with the irradiation experiment results and indicating that most equinox expression at 6 hpa wounds is in the

epidermis (Fig. 3d). Rare equinox+ muscle cells were also present at 6 hpa (equinox+colF-2+, Supplementary Fig. 7a). Furthermore, equinox expression at 6 hpa wounds was not explained simply by migration of pre-existing equinox+ DVB cells, because animals that had the DVB surgically removed still expressed equinox (Supplementary Fig. 7b). Extracellular regulated kinase (Erk) signaling mediates wound signaling in regeneration in several organisms, including planarians[48–51]. Phosphorylation of Erk occurs within minutes of injury in planarians[50], and inhibition of this process, using the inhibitor PD0325901 (PD), inhibited equinox expression at both 6 and 18 hpa (Supplementary Fig. 7c).

Bmp signaling controls the pattern of the planarian DV axis; bmp4 RNAi animals gradually lose dorsal gene expression and dorsal cell types during cell turnover after initiation of RNAi[17–19,45] (Supplementary Fig. 1a). Epidermal progenitors express dorsal and ventral markers from the stem cell stage to differentiation, and inhibition of bmp4 causes rapid dorsal emergence of ventral epidermal progenitors[45]. Accordingly, markers for dorsal epidermis and dorsal epidermal progenitors displayed reduced levels in bmp4 RNAi animals (Fig. 3e and Supplementary Fig. 8a, b) and normal expression of equinox at the dorsal side of the DVB was dependent on Bmp signaling (Fig. 1h). Moreover, equinox expression in the epidermal lineage at 16 hpa was enriched in dorsal over ventral epidermal progenitors (Fig. 3e and Supplementary Fig. 8c, d). These expression features help explain the requirement of Bmp signaling for the expression of equinox in regeneration. If equinox expression in the

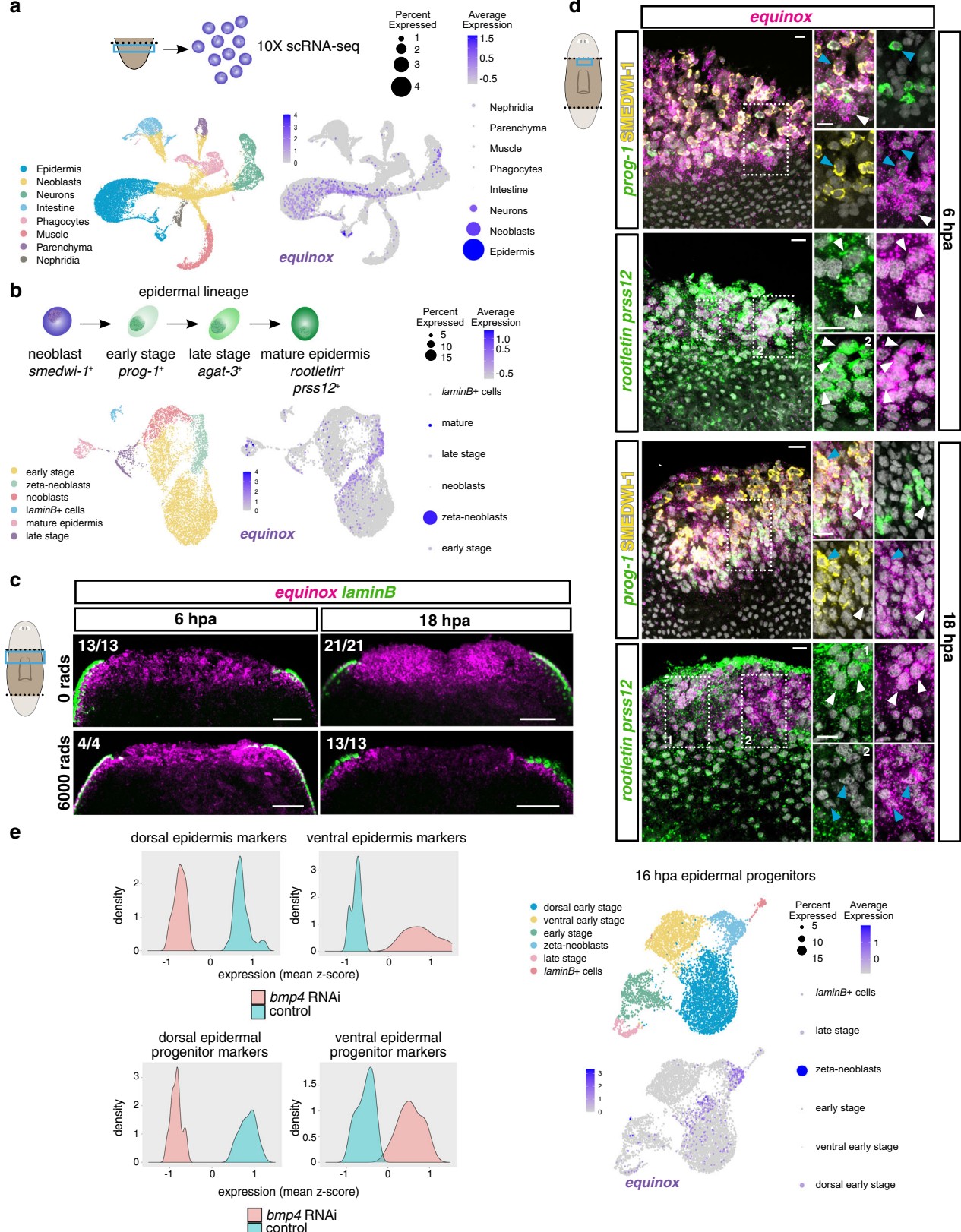

**Fig. 3 *equinox* is expressed in the planarian wound epidermis. a** Schematic of the 10X scRNA-seq experiment, UMAP and dot plots show *equinox* expression in the epidermis and in neoblast/neoblast progeny. **b** UMAP and dot plots show *equinox* expression throughout the epidermal lineage. **c** *equinox* wound-induced expression is mostly neoblast independent at 6 but not at 18 hpa. **d** Dorsal view of wound sites show *equinox* expression mostly in mature epidermis at 6 hpa and in epidermal progenitors at 18 hpa. Blue arrows, epidermal progenitors. White arrows, mature epidermis. **e** Histograms show downregulation of dorsal epidermis markers and dorsal epidermal progenitor markers after *bmp4* RNAi. UMAP and dot plots show *equinox* expression is enriched in dorsal epidermal progenitors at 16 hpa. Colored box, area depicted in pictures. Data shown in (**c**, **d**) are representative of four independent experiments. Scale bars, 100 μm (**c**) and 10 μm (**d**).

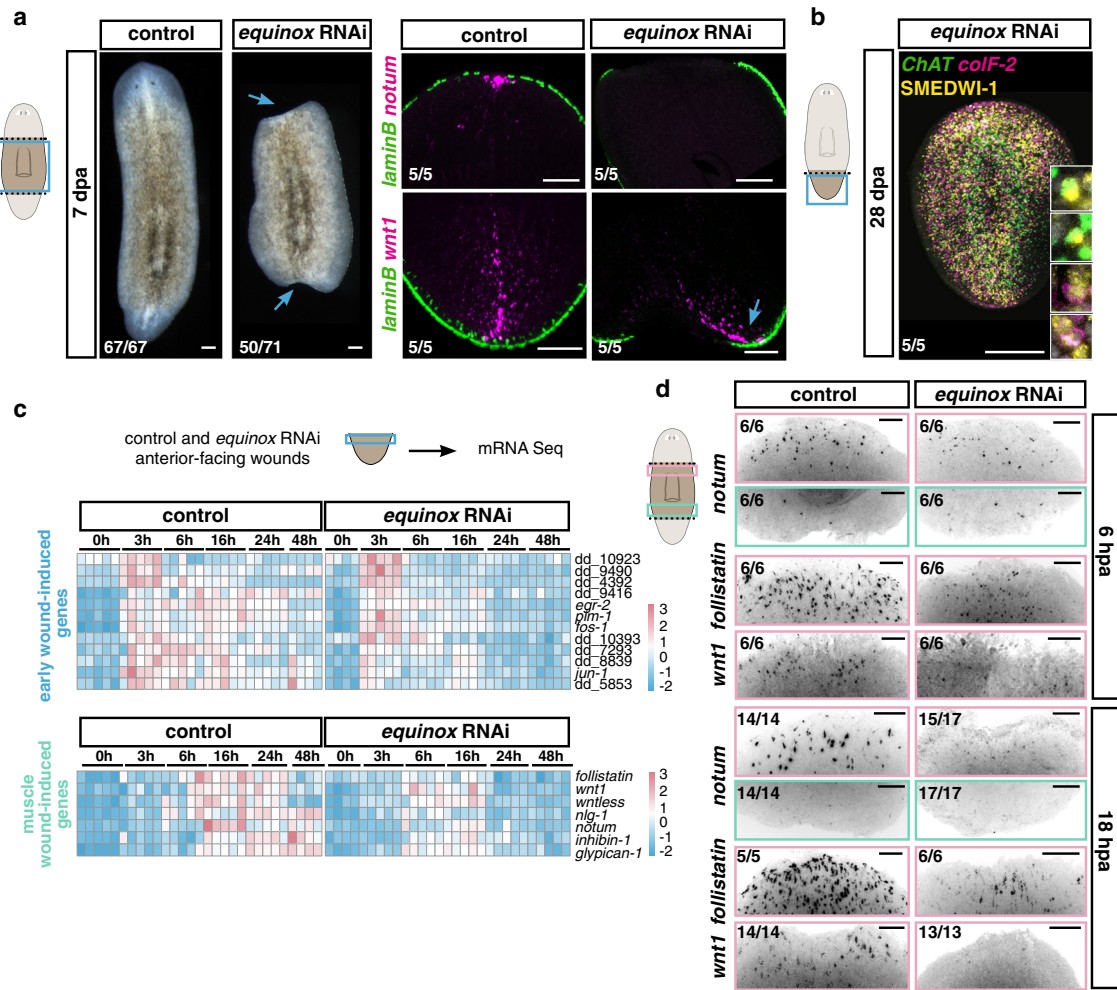

**Fig. 4 equinox is required for planarian regeneration. a** *equinox* RNAi animals do not regenerate. Blue arrows, no blastema (left) and asymmetric posterior pole (right). $n = 7$ independent experiments. **b** Normal tissue turnover in an amputated *equinox* RNAi fragment that failed to regenerate. $n = 2$ independent experiments. **c** Heatmap shows activation of early wound-induced gene expression but no maintenance of muscle wound-induced gene expression. **d** Validation by FISH. FISH results are representative of at least two independent experiments. Colored boxes, area depicted in pictures. Scale bars, 100 μm.

wound epidermis is associated with dorsal epidermal cells and/or cellular interactions between dorsal and ventral epidermis covering the wound, this could therefore be lost in *bmp4* RNAi animals.

**equinox is required for regeneration.** *equinox* RNAi animals were unable to regenerate following amputation (Fig. 4a). Blastema formation failed, and new tissues such as *laminB*+ epidermal DVB cells, anterior poles, brain, and eyes, failed to regenerate at anterior-facing wounds of both trunk and tail fragments (Fig. 4a and Supplementary Fig. 9a, b). At posterior-facing wounds, posterior poles were able to form in head and trunk fragments. However, reminiscent of the *bmp* RNAi phenotype, in most cases these poles formed asymmetrically at the old DVB and failed to display normal blastema growth (Fig. 4a and Supplementary Fig. 9a, b). Head fragments that formed a posterior pole also regenerated a pharynx in the pre-existing tissue and expressed posterior PCGs (Supplementary Fig. 9a). On the other hand, tail fragments always failed to regenerate anterior poles, did not express anterior PCGs, and were unable to form pharynges in the old tissue (Supplementary Fig. 9a). Anterior PCG expression domains also failed to regenerate in trunk fragments (Supplementary Fig. 9b). These defects indicated that *equinox* RNAi caused an even stronger block of regeneration than did *bmp4* RNAi, perhaps because direct RNAi of the gene caused

greater *equinox* inhibition. Anterior blastemas did not grow even after 14 dpa in *equinox* RNAi animals (Supplementary Fig. 10a). In trunk fragments, morphallaxis of posterior regions associated with newly formed posterior poles at the pre-existing DVB was sometimes observed at this later time point (Supplementary Fig. 10a).

The failure of regeneration in *equinox* RNAi animals was not a consequence of a lack of neoblast maintenance and differentiation capacity, because neoblasts of tail fragments that did not regenerate by 28 dpa were still able to differentiate into tail neurons and muscle cells (Fig. 4b and Supplementary Fig. 10b); i.e., neoblasts were capable of making pre-existing tissues but did not regenerate those that were missing. Furthermore, epidermis at the wound site of *equinox* RNAi animals was detected similarly to control animals (Supplementary Fig. 10c) and we did not observe any obvious defect in tissue turnover and viability in uninjured *equinox* RNAi animals (Supplementary Fig. 10d). These observations are consistent with the possibility that regeneration failed in *equinox* RNAi animals as a consequence of a regeneration-specific defect, as opposed to a problem in a process required generically for new cell-type production.

**Inhibition of equinox affects regeneration initiation.** To assess which step(s) in regeneration failed in *equinox* RNAi animals, we

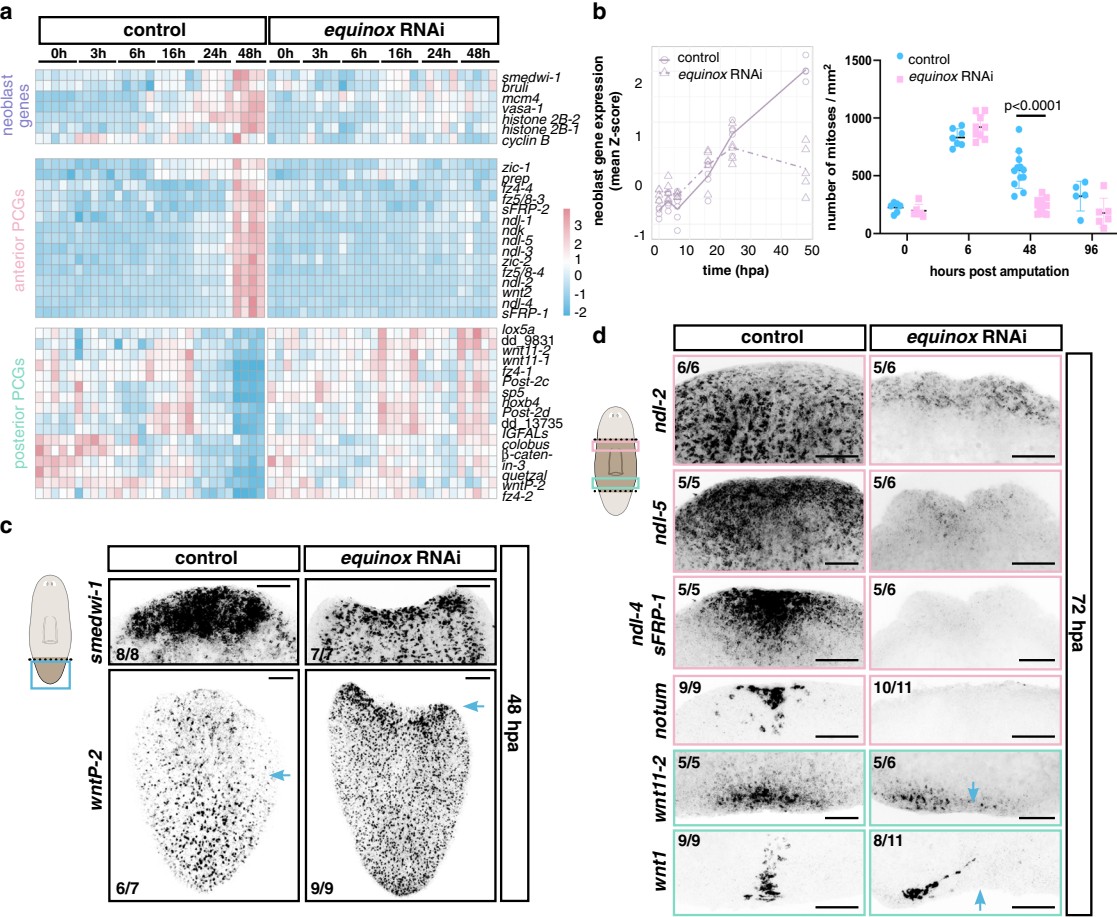

**Fig. 5 *equinox* is required for PCG rescaling and neoblast proliferation at wounds. a** Heatmap shows reduced expression of neoblast markers and anterior PCGs and no change of posterior PCG expression at anterior-facing wounds after *equinox* RNAi. **b** Reduced expression of neoblast-specific genes (left) and reduced numbers of mitotic cells (right) at *equinox* RNAi anterior-facing wounds. For phosphorylated histone H3 (H3P) labelings, *n* = 6 independent animals for control 0 hpa, *n* = 7 for control 6 hpa, *n* = 12 for control 48 hpa, and *n* = 5 for control 96 hpa; *n* = 6 independent animals for *equinox* RNAi 0 hpa, *n* = 9 for *equinox* RNAi 6 hpa, *n* = 12 for *equinox* RNAi 48 hpa, and *n* = 6 for *equinox* RNAi 96 hpa over two independent experiments. Data are presented as mean ± SD and analyzed with unpaired two-tailed Student's *t* test. **c** No neoblast wound accumulation or rescaling of posterior PCG expression (*wntP-2*) after *equinox* RNAi. *n* = 3 independent experiments. Blue arrows, anterior edge of PCG expression. (**d**) No anterior PCG expression (*ndl-2*, *ndl-5*, *ndl-4*, *sFRP-1*) or anterior pole formation (*notum*), and asymmetric posterior pole formation (*wnt1* and *wnt11-2*) after *equinox* RNAi. Results shown are representative from at least two independent experiments. Blue arrows, midline. Colored boxes, area depicted in pictures. Source data are provided as a Source Data file. Scale bars, 100 μm.

collected anterior-facing wounds of tail fragments at different time points following wounding for bulk RNA sequencing (Supplementary Data 3). *equinox* RNAi animals did not show signs of ventralization (Supplementary Fig. 10e), in contrast to *bmp4* RNAi animals, indicating *equinox* is not overtly involved in DV-axis patterning. *equinox* RNAi animals initiated an early wound response following injury (3–6 hpa, Fig. 4c and Supplementary Fig. 11). However, this response was less robust and did not persist at later time points (16–48 hpa, Fig. 4d). By FISH, muscle wound-induced gene expression was strongly affected in *equinox* RNAi (Fig. 4d and Supplementary Fig. 11). The neoblast gene expression signature failed to amplify at wounds at 24 and 48 hpa, indicating lack of neoblast accumulation and proliferation in *equinox* RNAi animals (Fig. 5a-c). Apoptosis was not clearly affected following *equinox* RNAi (Supplementary Fig. 12a). In addition, RNA-sequencing data showed that positional information in *equinox* RNAi animals did not reset; anterior PCGs were not induced in tail fragments, and posterior PCGs did not reduce expression at wounds (Fig. 5a). FISH further validated these observations (Fig. 5c, d and Supplementary Fig. 12b, c) and showed that these defects also occurred in trunk fragments

(Fig. 5d). *equinox* RNAi fragments failed to specify eye progenitors from neoblasts after head amputation, indicating stem cells failed to adopt fates for missing cell types after amputation (Supplementary Fig. 12d). These data collectively showed that regeneration of positional information and the associated regeneration of new tissues required *equinox* function. These results suggest that *equinox* has a specific role in initiating regeneration, involving maintaining a wound response program that promotes regeneration of positional information after injury.

**equinox is required for blastema outgrowth.** Short-term inhibition of *bmp4* did not result in severe ventralization[45], but significantly decreased *equinox*[+] expression at 18 hpa (Supplementary Fig. 13a). Under these conditions, all head and tail fragments were able to regenerate new tissues (pharynges and eyes) but only within the preexisting tissues (Fig. 6a). Expression of wound-induced genes (Supplementary Fig. 13b), accumulation of neoblasts at wounds, and rescaling of positional information (Fig. 6b) all occurred after short-term *bmp4* RNAi. However, no blastema formation was observed in any regenerating fragment. These

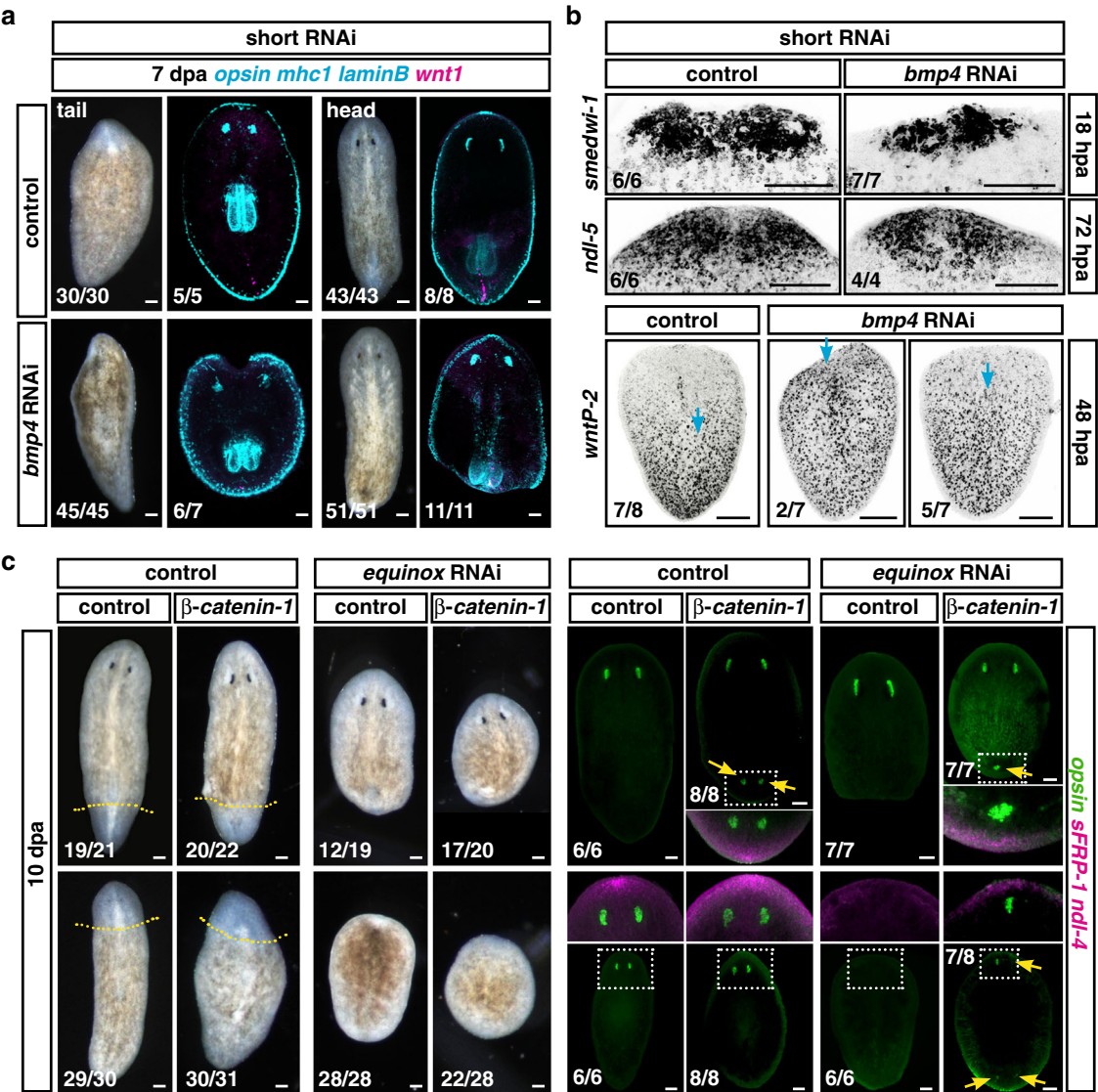

**Fig. 6 *equinox* is required for blastema formation. a**, **b** Short-term *bmp4* RNAi is sufficient to block blastema formation (**a**) but does not block neoblast accumulation or PCG rescaling (**b**). Blue arrows, anterior edge of posterior PCG expression. **a**, **b** *n* = 2 independent experiments. **c** The *equinox* RNAi regeneration defect is not suppressed by β-*catenin-1* RNAi. *equinox; β-catenin-1* double RNAi animals were able to rescale positional information and form new photoreceptor neurons (*opsin*) without growing blastemas. *n* = 3 independent experiments. Scale bars, 100 μm.

results, with short RNAi conditions, suggest that *bmp4* is required for blastema growth itself and that this process can be decoupled from other hallmarks of regeneration. These observations prompted us to consider whether *equinox* has a role in blastema growth in addition to and independent from its requirement for positional information regeneration.

Both *myoD* and *follistatin* RNAi regeneration failure phenotypes can be rescued by inhibiting β-*catenin-1*, suggesting that blastema formation failure in those scenarios is a consequence of defects in positional information re-setting[35,38]. By contrast, double *equinox; β-catenin-1* RNAi animals still failed to form a blastema (Fig. 6c and Supplementary Fig. 13c, d), indicating that lack of blastema formation could not readily be explained only by defects in the level of Wnt signaling at wounds associated with PCG resetting. Moreover, FISH experiments showed expression of anterior PCGs and presence of anterior tissues (i.e., photoreceptor neurons) at anterior and posterior-facing wounds in *equinox; β-catenin-1* double RNAi animals lacking blastema outgrowth, further demonstrating that changing the positional

information environment of the wound in *equinox* RNAi animals was insufficient to lead to blastema formation (Fig. 6c). Similarly, inhibition of Erk signaling also blocked head blastema formation in β-*catenin-1* RNAi animals, even though under these conditions some expression of anterior positional information and differentiation of anterior tissues within the old tissues was observed (Supplementary Fig. 13e).

Blastema growth is a hallmark of regeneration. However, not all injuries require blastema formation for their repair. For example, incisions or small tissue removal can heal and regenerate through a tissue turnover-mediated process[24]. We therefore examined whether *equinox* was required for all types of regeneration or specifically for contexts requiring blastema formation for regeneration. *equinox* RNAi animals were able to regenerate eyes after eye resection or after a small wedge that was able to heal (Fig. 7a and Supplementary Fig. 14). Inhibition of *equinox* did not affect pharynx regeneration after pharynx removal through a small dorsal incision (Fig. 7a and Supplementary Fig. 14). However, *equinox* RNAi animals were incapable

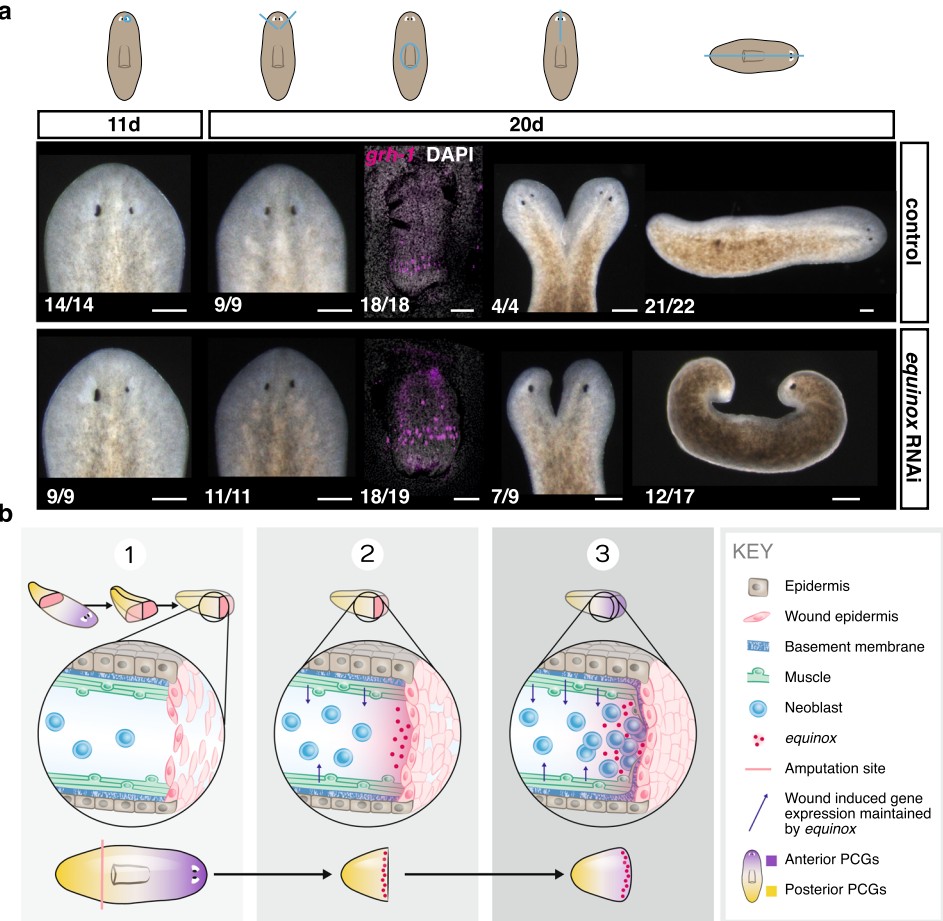

**Fig. 7 equinox is required for regeneration contexts that need blastema outgrowth. a** equinox is required for regeneration through blastema formation. $n = 3$ independent experiments for sagittal amputations, $n = 2$ independent experiments for all other injuries. Blue lines, injury. Scale bars, 100 μm. **b** Model: (1) *wound healing* involves formation of a wound epidermis lacking basement membrane separation from underlying tissues. (2) *Wound signaling* is promoted and sustained by equinox expression in the wound epidermis. (3) *Positional information* regeneration is promoted by wound signaling and leads to specification of missing cell types in proliferating neoblasts at wounds. These steps promote initiation of blastema formation in regeneration.

of regenerating from multiple wound types that required blastema formation at the amputation site, such as after sagittal amputations, or midline incisions that were prevented from fusing (resulting in two half-heads and one body) (Fig. 7a and Supplementary Fig. 14). These findings suggest that *equinox* is only required for contexts where resetting positional information and blastema growth occur for normal regeneration.

In conclusion, the previously uncharacterized gene *equinox* encodes a predicted extracellular protein conserved in many animal phyla that is expressed in the planarian wound epidermis and is required for regeneration. *equinox* is required for other cell types, including muscle and neoblasts, to maintain expression of their wound-response programs following injury (Fig. 7b). Sustained expression of wound-induced genes is associated with resetting of positional information (facilitated by modulation of Wnt signaling activity[38]) and proliferation of neoblasts (likely activated by Follistatin[36,38]) at wounds. The resetting of positional information is required for the specification of missing cell types and regeneration. Consequently, these processes required *equinox*, especially in the context of anterior regeneration. We also propose that in addition to positional information resetting, *equinox* action in the wound epidermis is essential for blastema outgrowth. We propose that *equinox* expression may be a critical signal in early epidermal-muscle cell communication following wounding, potentially facilitated by the absence of basement membrane

separating the wound epidermis from underlying cells at the wound. These findings reveal an important role of the wound epidermis in planarians, and might help to understand the necessity of this tissue for the broad regenerative potential of diverse organisms.

## Methods

**Animal husbandry**. *Schmidtea mediterranea* clonal asexual strain CIW4 animals, starved for 7–14 days prior to experimentation, were used for all experiments. All animals utilized were healthy, not previously used in other procedures, and were of wild-type genotype. Animals were cultured in plastic containers or petri dishes for experiments, in 1x Montjuic water (1.6 mmol/l NaCl, 1.0 mmol/l CaCl$_2$, 1.0 mmol/l MgSO$_4$, 0.1 mmol/l MgCl$_2$, 0.1 mmol/l KCl and 1.2 mmol/l NaHCO$_3$ prepared in Milli-Q water) at 20 °C in the dark. Animals were fed blended calf liver once a week and cleaned twice weekly.

**Replication, size estimation, and randomization**. Number of independent experiments performed are indicated in figure legends; numbers of animals used in each experiment are indicated in each panel. No sample size estimation was performed. Animals for all experiments were randomly selected from a large collection of clonal animals. All animals were included in statistical analyses, without exclusions. Images were not randomized.

**Bulk RNA sequencing and analysis**. RNA sequencing data from *follistatin* and *myoD* RNAi tail regeneration was analyzed from previously published experiments ([35], GEO: GSE99067). For *bmp4* and *equinox* RNAi samples, animal fragments were placed in Trizol (Life Technologies), and then frozen at −80 °C. Upon thawing, fragments were resuspended in Trizol by pipetting with a p1000 until dissolved. Total RNA was isolated using Trizol standard isolation. Libraries were

prepared using the Kapa HyperPrep mRNA-Seq Kit Illumina Platform, and barcoded with Kapa dual-indexed adapters (KapaBiosystems). A second 0.9× bead clean-up was performed to purify libraries. Libraries were sequenced on an Illumina Hi-Seq 2500 with 50 bp reads. Reads were mapped to the dd_Smed_v6 transcriptome[52] (http://planmine.mpi-cbg.de/planmine/begin.do) using bowtie v1.1.2[53] with -best alignment parameter. The number of mapped reads per contig in every cell was quantified using the coverageBed utility from the bedtools v2.26.0 suite. Reads from the same isotigs were then summed to generate raw read counts for each transcript.

**10X single-cell mRNA sequencing and analysis**. Cell samples were collected from five planarian conditions with tissue from the same post-pharyngeal region; (1) 0 hpa, (2) 6 hpa anterior-facing wounds, (3) 16 hpa with anterior-facing wounds, (4) 16 hpa with posterior-facing wounds, (5) 16 hpa anterior-facing wound with 4 feedings of control dsRNA over 21 days. Sample 5 was included as a replicate of this time point to increase the number of cells, and therefore increasing the power of this dataset. Fragments from ~30 large animals were collected for each condition. Each condition was run as a single sample in 10X library preparation. Tissue fragments were gently amputated for dissociation using a scalpel, and collected in CMFB [calcium–magnesium free solution with 1% BSA (400 mg/L NaH$_2$PO$_4$, 800 mg/L NaCl, 1200 mg/L KCl, 800 mg/L NaHCO$_3$, 240 mg/L glucose, 1% BSA, 15 mM HEPES, pH7.3)]. Pieces were vigorously pipetted in 50 ml of CMFB with 1.0 mg/mL collagenase as described[43]. Cell suspensions were passed through a 40 μm filter and centrifuged at 300 g for 5 min. Isolated cell suspensions were incubated in Hoechst 33342 (40 μl/ml) for 45 min in the dark. For samples 1–4 propidium iodide (PI) (3 μl/ml, Sigma-Aldrich) was added after Hoechst incubation, for sample 2 calcein (1 μl/ml, ThermoFisher) was added to samples after Hoechst incubation for 10 min. Cells were sorted based on DNA content and PI negative, X-insensitive cells were recovered[54]. Cells with intermediate calcein content were permitted for sample 2. At this early wounded time point, calcein intermediate cells were allowed assuming some wounded cells might have a compromised membrane. Likely for this reason the overall quality of lane 2 cells was lower (see Supplementary Fig. 5). Planarian cells were sorted into CMFB (1% BSA), spun down at 500 rpm for 10 min, and resuspended in CMFB (1% BSA) at a predicted density of 700 cells/μl according to the FACS sorting count. Cells were then processed by the WIGTC core (Whitehead Institute) using 10X Genomics Chromium Controller and Chromium single cell 3′ library & Gel Bead Kit (PN 1000006) following standard manufacturer's protocol. An estimated 16,000 cells were loaded per sample to obtain the maximum number of cells recommended per sample. Amplified cDNA libraries were quantified using a bioanalyzer, and size selected using magnetic beads according to manufacturer recommendations. Samples were sequenced on an Illumina HiSeq 2500 (28 × 40 paired-end reads) across five lanes. Sequencing reads were mapped using a GTF file of Smed_v6 genes in the context of the Smes_g4 genome. This GTF file was generated by using BLAT to map all Smed_v6 transcripts to the Smes_g4 genome and assigning each transcript to a single genome location based on the best alignment score. Transcripts were then collapsed using genome location prior to mapping using the Cell Ranger pipeline. Cells were assessed for nUMI, nGene and percent mitochondrial transcript content, which was represented in violin plots. Percent mitochondrial content was based on 10/15 mitochondrial genes reported in ref. [55] which are represented in v_6 of the Dresden transcriptome (dd_Smed_v6_258_0_1, dd_Smed_v6_289_0_1, dd_Smed_v6_292_0_1, dd_Smed_v6_297_0_1, dd_Smed_v6_344_0_1, dd_Smed_v6_505_0_1, dd_Smed_v6_753_0_1, dd_Smed_v6_957_0_1). Any cells with nFeature_RNA < 200 or nFeature_RNA > 2500, were removed from the dataset prior to analysis. 10× analysis was performed using Seurat v3.02 where cells were visualized using the uniform manifold approximation and projection (UMAP) algorithm. CCA-based integration was performed for the plots that contained all tissues and all epidermal lineage cells to address any potential batch effects between lanes. CCA-based integration was omitted for 16 h epidermal lineage plots because it removed known validated clusters (i.e. laminB+ cells and late-stage progenitor cells). Dimensions used and resolution of the plots are as follows: (1) all tissues (11, 1), (2) epidermal lineage (14, 0.5), (3) 16 h epidermal lineage (14, 0.3).

**Protein domain analysis**. Reciprocal blastx/tblastn with specific species in NCBI and the longest predicted ORF of the equinox gene (dd_Smed_v6_20318_0_1) from the dd_v6 transcriptome assembly was performed. Proteins with reciprocal best blast hits to equinox were inspected for domain architecture similarities using SMART with HMMER searches of Outlier homologs, PFAM domains, and signal peptide prediction. The protein domain structure of Equinox and VWDE predicted by SMART is graphically represented. showing predicted signal peptide, TSP1, EGF, and EGF-like domains. Phyla containing the same gene structure of one or two N-terminal TSP1 domains, followed by several EGF and EGF-like domains were annotated on the species tree. Another secreted protein class displayed substantial similarity to the C-terminus of Equinox; representatives of this class including in vertebrates lack the TSP1 repeats and have a von Willebrand Factor D domain.

**Phylogenetic analysis**. equinox, vwde, and teneurin genes were identified from multiple species by BLAST to Schmidtea mediterranea Equinox protein sequences

using only the EGF domain containing region. Multiple sequence alignment of protein sequences was performed using MUSCLE to the EGF domain-containing region of Schmidtea mediterranea Equinox. Sequences were then trimmed by hand to remove areas that did not align with the EGF domain-containing region of Schmidtea mediterranea equinox. Sequences were then further trimmed with Gblocks using the following parameters; -b3 = 15 -b4 = 2 -b5 = a -e = .gb -p = t -g. Phylogenetic trees were constructed using Bayesian inference (MrBayes v3.2.2 ×64). The analysis was performed using two independent runs with four chains each for 1 million generations or more, and 25% of trees were discarded as burn-in. All protein sequences used for analysis with accession numbers and the nexus file (.nex) on which the tree is built are provided in Supplementary Data 2.

**Gene cloning**. equinox was cloned using the following primers: fwd: 5′ gggccagt-tacttcacaagc; rv: 5′ gagccagagaaagattgcgg; bmp4 (dd_Smed_v6_17402_0_1) accession number: GenBank ABV04322.1. All constructs were cloned from cDNA into the pGEM vector (Promega). These constructs were used to synthesize RNA probes and dsRNA for RNAi experiments.

**RNAi**. For RNAi experiments, dsRNA was synthesized by in vitro transcription reactions (Promega) using PCR-generated templates with flanking T7 promoters, followed by ethanol precipitation, and annealed after resuspension in water. The concentration of dsRNA varied in each prep between 4 and 7 μg/ml. dsRNA was then mixed with planarian food (liver)[35] and 2 μl of this mixture per animal (liver containing dsRNA) was used for feedings. This protocol yields stronger gene inhibition than the RNAi feeding protocol that involved agarose and bacteria-based dsRNA production in[17]. bmp4 RNAi animals (Fig. 1; Supplementary Figs. 1–3, Supplementary Fig. 8) were fed four times in two weeks. Animals were then amputated seven days after last feeding. Short term bmp4 RNAi animals (Fig. 6; Supplementary Fig. 13) were fed once and animals were amputated 7 days after this feeding. equinox RNAi animals (Figs. 4, 5 and 7; Supplementary Figs. 9–12, 14) were fed between six and ten times in 3–5 weeks and amputated 3 days after last feeding. equinox RNAi animals for homeostasis experiments (Supplementary Fig. 10) were fed 12 times during 10–12 weeks. Double equinox; β-catenin-1 RNAi animals (Fig. 6; Supplementary Fig. 13) were fed eight times in 4 weeks with equinox dsRNA, and then two times in 2 weeks with a 1:1 mix of equinox: β-catenin-1 dsRNA. Animals were amputated a day after last feeding. β-catenin-1 RNAi animals incubated with DMSO or the Erk inhibitor were fed once (Supplementary Fig. 13). For regeneration experiments, animals were amputated into three pieces (head, trunk, and tail pieces). Fragments were scored and fixed for further analysis at different timepoints following amputation.

**Fluorescence in situ hybridizations, immunostainings, and HE stainings**. RNA probes were synthesized and whole-mount FISH was performed[35]. Briefly, animals were killed in 5% NAC and treated with proteinase K (2 μg/ml). Following overnight hybridizations, samples were washed twice in pre-hybridization buffer, 1:1 pre-hybridization-2× SSC, 2× SSC, 0.2× SSC, PBS with Triton-X (PBST). Subsequently, blocking was performed in 10% Western Blocking Reagent (Roche, 11921673001) PBST solution for DIG probes, or in 5% Horse serum and 5% casein for DNP and FITC probes. Antibody washes were then performed for one hour followed by tyramide development. Peroxidase inactivation with 1% sodium azide was done for 90 min at room temperature. SMEDWI-1 antibody labeling was then performed with a 1:1000 dilution of primary antibody in 10% Western Blocking Reagent (Roche, 11921673001). SMEDWI-1 antibody presence was detected using tyramide development as above. Brightfield images were taken with a Zeiss Discovery Microscope. Fluorescent images were taken with a Leica SP8 confocal microscope. Co-localization analyses of FISH signals were performed using Fiji/ImageJ. For each channel, histograms of fluorescence intensity were used to determine the cut-off between signal and background. All FISH images shown are representative of all images taken in each condition, and are maximal intensity projections, except otherwise indicated. All images, unless otherwise indicated, are anterior up. Animals for FISH experiments of early timepoints following amputation (6 hpa) were treated with 5% NAC for only 2 min (instead of 3–5) to better preserve wound integrity. For H&E stainings, animals were fixed in the same way as for FISH experiments, and mounted in Histogel specimen processing gel (Thermo Scientific). Samples were then placed on ice for 10 min to solidify. Cassettes with embedded animals were placed in 70% ethanol and submitted to the Histology core at the Koch Institute for integrative cancer research (MIT) for sectioning and staining.

**TUNEL and H3P labeling**. For both TUNEL and H3P labeling, fixed animals were bleached overnight at room temperature in H$_2$O$_2$ (Sigma, 6% in 1×PBSTx), incubated 10 min in Proteinase K solution (2 μg/ml in 1×PBSTx with 0.1% SDS) and post-fixed in formaldehyde (4% in 1×PBSTx). For H3P labeling, animals were then incubated overnight at room temperature in anti-phospho-Histone H3 antibody (Millipore 05-817R-I, clone 63-1C-8; 1:300 in 5% inactivated horse serum) Samples were washed with PBSTx, then placed in goat anti-rabbit antibody (ThermoFisher 65-6120, 1:500 in 5% inactivated horse serum) overnight at room temperature. After PBSTx washes, samples were developed in fluorescein tyramide (1:3000 in PBSTx, with 0.003% H$_2$O$_2$) for 10 min at room temperature. Samples

were washed in PBSTx and labeled with DAPI (Sigma, 1 μg/ml in PBSTx) before mounting. For TUNEL, single animals were transferred to a 96-well U-bottom plate. PBSTx was replaced with a 20 μL reaction mix (3 parts ApopTag TdT enzyme mix, 7 parts ApopTag reaction buffer), and incubated overnight at 37 °C. Animals were then washed in PBSTx followed by development in a 20 μL development solution (1 part blocking solution, 1 part ApopTag anti-digoxigenin rhodamine conjugate), and incubated in the dark at room temperature overnight. Samples were washed in PBSTx and counterstained with DAPI (Sigma, 1 μg/ml in PBSTx). TUNEL was performed using reagents from the ApopTag Red in Situ Apoptosis Detection Kit (Millipore, #S7165).

**Erk signaling inhibition**. PD0325901 (in short, PD) was dissolved in DMSO, used at 10 μM, and replaced daily. Animals were incubated in PD one day prior to amputation, amputated, and fixed after 6 and 18 h to assess *equinox* expression. For RNAi experiments, animals were fed *β-catenin-1* dsRNA, immediately placed in PD, and amputated later the same day. Fragments were fixed for further analysis 10 days after amputation.

**Irradiation**. Animals were irradiated using a dual Gammacell-40 cesium source set to deliver 6000 rads. Animals were amputated 4 days after irradiation.

**Surgical procedures**. For all surgical procedures, animals were placed on moist filter paper on a cold block to limit movement. In order to selectively resect eyes, the tip of a microsurgical blade was used to remove eyes. Pharynx resection was performed by surgical extraction through a small longitudinal dorsal incision. Head wedges were made by performing cuts at the edge of each eye. Sagittal cuts were performed at the midline. Midline incisions were performed by making a medial incision from the head tip to the anterior boundary of the pharynx. Incised animals were immobilized using Type IV, 5% ultra-low melting agarose (Sigma) and solidified gel was covered with filter WhatmanTM paper (GE Healthcare, Life Sciences) soaked in Holtfreter's Solution. Animals were left at room temperature overnight, recovered by cutting the surrounding gel, and placed in planarian water (Fig. 7 and Supplementary Fig. 14). AP1 cuts were made immediately posterior to auricles (Supplementary Fig. 3).

**Quantifications and statistical analysis**. Pairwise differential expression analysis was performed using DESeq[56]. Expression values from DESeq normalization were scaled, row-wise, to generate z-scores for heatmaps and visualized using the pheatmap package, or the ggplot2 package with geom_density and geom_point functions (https://ggplot2.tidyverse.org). Significance is reported as padj values, with padj < 0.05 used as a cutoff. For TUNEL and H3P stainings, comparisons between two groups were done using unpaired Student's *t* test analyses (Prism software). Graphs show mean and standard deviation.

**Reporting summary**. Further information on research design is available in the Nature Research Reporting Summary linked to this article.

## Data availability
The sequencing data generated in this study have been deposited in the GenBank database under accession codes (1) GSE179290 [Single cell gene expression profiling using 10x v3, (2) GSE179291 [bmp4 RNAi gene expression profiling; (3) GSE179293 [*equinox* RNAi gene expression profiling; (4) OM864265 (*equinox* gene deposition). The anti-H3P labeling data and processed DEseq comparison data for bulk sequencing generated in this study are provided in the Supplementary Information/Source Data file.

## Code availability
All code for data cleaning and analysis associated with the current submission is available through the following resources; (1) Seurat and CCA-based integration: (2) DESeq: (3) Cell Ranger.

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

## Acknowledgements

The authors thank Shannon Moreno for collection of single cells, George Bell for 10x mapping, Isaac Oderberg for discussions, and Kwadwo Owusu-Boaitey for embedding animals for histology, and Caitlin Rausch for the model illustration. We thank the Eleanor Schwartz Charitable Foundation for support. P.W.R. is an investigator of HHMI and an associate member of the Broad Institute. P.W.R. acknowledges NIH support (R01GM080639).

## Author contributions

M.L.S., J.K.C., P.W.R. designed the study; M.L.S., J.K.C., and C.L.M. carried out experiments; M.L.S., J.K.C., and C.L.M. analyzed data; and M.L.S., J.K.C., and P.W.R. wrote the manuscript.

## Competing interests

The authors declare no competing interests.
