## [Peer Review File · Nature Communications]

The planarian wound epidermis gene equinox is required for blastema formation in regenerationREVIEWER COMMENTS

Reviewer #1 (Remarks to the Author):

In the manuscript 'The planarian wound epidermis gene *equinox* promotes positional information resetting to initiate regeneration' Scimone, Cloutier et al. identify a new gene, *equinox*, which could be a target of *bmp4* in promoting the regenerative response. *equinox* RNAi planarians fail to reestablish positional information, to upregulate proliferation and do not regenerate a blastema. The authors propose that *equinox* could be an early signal in epidermal-muscle cell communication following wounding in planarians, similar to the role of the wound epidermis seen in vertebrates. The finding and characterization of this new gene is relevant in the field of regeneration and biology in general. The work is of high scientific quality. However, the interpretation of some results is questionable. Before publication, the following concerns must be addressed.

Major concerns:

1. The main concern of the presented study is the assumption that the inability to regenerate of *bmp4* and *equinox* RNAi animals results from the inability to reset positional information, which in fact is the title of the manuscript, and should be revised.

According to the results, *bmp4* and *equinox* RNAis do not only show problems in positional information but show a drastic decrease in the number of neoblasts near the wound during regeneration and a decrease in neoblast-associated genes. Although the authors argue that this is not the reason of the non-regenerating phenotype, an analysis of the dynamics of proliferation during the early and late regenerative phases is required, to test whether *bmp4* and *equinox* RNAis show a decreased or a delayed mitotic response. It could be that the inability to regenerate is linked to a problem in triggering the regenerative mitotic response.

The arguments of the authors to claim that the proliferative response is not a cause of the non-regenerative phenotype are confusing. For instances, in Lines 253-257 it is read: 'The failure of regeneration in *equinox* RNAi animals was not a consequence of a lack of neoblast maintenance and differentiation capacity, because neoblasts of tail fragments that did not regenerate by 28 dpa were still able to differentiate into tail neurons and muscle cells (Fig. 3b), i.e., neoblasts were capable of making pre-existing tissues but did not regenerate those that were missing.' This argument is not valid, since it could be that *equinox* is not required for neoblast maintenance and differentiation, and that's the reason why some structures are differentiated several days after the cut, but it could be required for the mitotic response of neoblasts at initial regeneration stages. This is indeed a main claim of the manuscript, that *equinox* has a specific role during regeneration but not in the maintenance of tissues.

Even more, the observation that in *bmp4* and *equinox* RNAis the brain, eyes, pharynx... can be regenerated in the pre-existent tissues suggests that the problem is not the positional information but the inability to create enough new cells to make a blastema where to differentiate the missing structures.

In the same direction, the observation that *bcat1* RNAi does not rescue the phenotype further suggests that the problem is not positional information but it could be proliferation. The same occurs with the results showing that small but not big injuries can be regenerated. For instances, an eye can be regenerated, but not a pharynx, which demands much more many cells, although neither of them is different in the requirement of a reset in the positional information.

2. The second main concern refers to whether the early wound response is affected in *bmp4* or *equinox* RNAis. The authors claim that early wound response takes place in *bmp4* or *equinox* RNAi. However, according to the RNA-seq data and to the in situ, the initial wound response in *bmp4* and *equinox* RNAi animals is not a normal response. In general, a decrease of the wound-induced genes is seen in the RNA-seq (fig 1b and 3c). In fact, *equinox* is chosen because it is downregulated during the first hours after *bmp4* RNAi, as also seems to occur with *inhibin*, *glypican* or *fos-1*. And, clearly, the pattern of expression of the wound genes at 6h is not the same in *bmp4* or *equinox* RNAi animals than in controls. Thus, according to the results showed, the early wound response is already affected in these phenotypes.

Furthermore, along the manuscript it is not clear whether the authors consider that *bmp4* and *equinox* exert a role during the first hours of regeneration or not. For instances, when comparing with *follistatin* or *myoD* RNAis they also conclude that 'the regeneration-specific defect in *bmp4* RNAi animals might therefore involve a previously uncharacterized process associated with regeneration initiation.' This aspect of the manuscript is very confusing and should be clarified. (An

advice would be also to omit the comparison with the follistatin or myoD RNAs, since it just increases complexity but not provides any answer to equinox function.)

3. In planarians the early wound response includes the expression of the wound-induced genes, the activation of mitosis and also the activation of apoptosis. In the previous point 1 the importance of analyzing mitosis has already been exposed. To clarify whether the early regenerative response takes place (previous point 2), the apoptotic response should be also analyzed. It is relevant to understand whether the non-regenerative phenotype described is associated to a failure in triggering the very early apoptotic response, directly associated to the injury, or to the late apoptotic response, associated to the integration of the new and the old tissue, which could be linked to the appearance of differentiated structures in the pre-existent tissue showed in the RNAs.

4. May be due to the compact form of the manuscript, there are some concepts that are not explained clear enough for common readers.

Line 44, an explanation of what is a blastema is required.

Lines 47-49 'Once the wound is covered, epithelial cells proliferate to form a thick epithelium, mobilization of progenitors occurs (involving tissue resident stem cells and/or dedifferentiation), and the regenerative blastema forms'. It is confusing if it refers to vertebrates, planarians... Lines 80-87, the introduction of PCGs is not clear enough for non-planariologists.

Lines 105-107, the concept of DV boundary is not properly explained.

5. Since equinox is a new gene, a better phylogenetic analysis is needed, including more species and showing the alignment and the tree. Some questions can be answered with it. For instances, do some species that have Thrombospondins also have Equinox, or they only have one of them at a time? Is it a correlation between having echinox and the regeneration capacity?

Regarding Thrombospondins, have them been related with BMP signal in any report?

6. Are *bmp4* and equinox co-expressed in the DV boundary? And in the blastemas? Is *bmp4* expressed in equinox RNAi animals? There are very direct questions that should be answered in the manuscript.

7. equinox seems to be a target of *bmp4*, and it shows a very specific dorsal localization. However, equinox RNAi animals do not show DV defects. Is equinox a *bmp4* mediator only in the regenerative response? And this response is not linked to the role of *bmp4* in polarity? A further discussion is required.

Minor concerns:

1. Scheme in figure 4- What are the pale pink cells in the wound that seem not to express echinox? According to figure 2c, echinox is expressed in several lines of cells in the wound, not only in the ones contacting the pre-existent tissue. It is confusing.

2. How can be explained the asymmetry of the poles when they can regenerate? Some comment about that would be helpful.

3. Figure 3b. Control is missing. It is necessary for readers.

4. Lines 118-120. 'Anterior poles formed more often at *bmp4* RNAi anterior-facing wounds in the animal anterior than in the posterior (e.g., trunk fragments, Extended Data Fig. 2c), but these wounds still lacked blastema outgrowth. Posterior poles formed frequently, even though no blastema was observed at *bmp4* RNAi posterior-facing wounds (Extended Data Fig. 2a, c)'. It is not clear what means posterior, what is more frequent... It should be clearly explained. Specifying that it is based in the expression of *notum* in A and *wnt1* in P will help.

5. It is claimed that the DV boundary can be recognized by the expression of *laminb*. However, according to image 2b, *laminb* seems to be ventral to the DV boundary.

Furthermore, it is very interesting that the expression of *laminb* and equinox seem to be separated by 1 cell. Is it always like that?

6. The expression of echinox in intact animals is clearly dorsal. Has it also a dorsal pattern in the regenerating wounds? In images in 2c it seems that it is both, in D and V, although the in situ and the SC-seq data shows that at 18h it is more dorsal than ventral. At some point during regeneration it must be relocated in the dorsal part. When does it happen? It could be that during the first hours equinox is delocalized and then some upstream signal restricts it to dorsal. This could be linked to the different role that it exerts while it is delocalized with respect to its localized expression in intact animals.

Reviewer #2 (Remarks to the Author):

In the manuscript, the authors present *equinox*, a gene encoding a likely secreted protein with several conserved domains, and a strong requirement for regeneration initiation in planarians. They identified *equinox* in a search for BMP4-dependent genes after they showed that BMP4 RNAi had more severe and general effects on planarian regeneration than previously published.

The authors convincingly show that *equinox* RNAi specifically blocks regeneration at an early stage, while it does not affect homeostatic turnover in non-injured planarians. They characterized its expression pattern and show it is activated in various cell types, predominantly in epidermis and epidermal progenitor cells. Based on the expression pattern, they hypothesize *equinox* may be involved in epidermis-muscle communication as part of a wound epidermis signaling program initiating regeneration.

This gene is truly interesting as it is relatively uncharacterized and has a very strong regeneration phenotype that is specific for regeneration initiation.

However, there are two major points I have to raise:

1) The authors state that *equinox* is not present in vertebrates. I did a quick Blast search with the *equinox* sequence, and found Van Willebrand Factor D and EGF Domains (VWDE), conserved in vertebrates. VWDE was recently identified by the Whited lab as a secreted protein induced in different cells in regeneration blastemas of several regeneration competent animals and required for limb regeneration in axolotl (<https://pubmed.ncbi.nlm.nih.gov/32163674/>).

As *equinox* and VWDE are best Blast hits in reciprocal Blast searches, they have a similar function in regeneration, and have a regeneration-associated expression pattern, it is very likely that *equinox* is a functional homolog of VWDE. It is important that the authors clarify the evolutionary relationship with VWDE and cite the study.

2) The study seems rather preliminary. Are there any transmembrane proteins in stem cells/muscle cells/epidermal cells or blastema cells that could act as receptors for *Equinox*? The scRNAseq dataset might reveal some of these signaling components. Given that *equinox/vwde*'s function in regeneration has been shown for other regenerating animals, identifying the receptor and downstream signaling pathways would add novelty to the study, especially, when it can be shown that these receptors or pathways are conserved in other regenerating animals.

Other points:

3) I find the BMP4 phenotype not convincing enough to speak of a general early role of *bmp4* in planarian regeneration. In Fig. 1, at posterior blastemas, there is not much difference in gene expression in control and *bmp4* RNAi animals. Is there a way to quantify the reduced expression of regeneration genes, or do the authors have RNAseq data on posterior blastemas similar to what they show for anterior blastemas in Fig.1b?

In Ext. Data 2e, it looks like there is quite a big blastema (unpigmented tissue) that even contains eye spots? The posterior-facing wound is more convincing here.

4) The authors tested a requirement for BMP4, follistatin and MyoD for *equinox* expression. Only *bmp4* RNAi had an effect. Since there is a very early and conserved requirement for ERK signaling in regeneration, yet the downstream effectors are not known, it would be interesting to see if *equinox* activation depends on this pathway.

5) To my unskilled eye, the H&E staining in Fig. 2 shows an open wound and does not seem to be covered by any epidermis. Staining with an epidermal marker would be helpful, or at least a close-up of the H&E images, so that epidermal cells can be identified by morphology.

6) Fig. 3c: *equinox* expression is induced during the first 3h after injury in *equinox* RNAi planarians. Do the authors have an explanation for this?

- 7) Fig. 1b: the time scale is missing on the x-axis.
- 8) No percentage of cells that express the gene in the different Seurat groups in fig. 2e, 2h.
- 9) On some occasions, FISH is used in the text to refer to what looks like non-fluorescent ISH to me (fig1c, 1d, 1e3d, 3e, 3f, 4b)
- 10) SMEDWI1 (Fig. 2g) is not mentioned in the main text.

Reviewer #3 (Remarks to the Author):

This paper investigates the consequences of high dose bmp4 knockdown and through a series of bioinformatic and wet lab experiments identifies a role for a novel wound epidermis gene, equinox, in blastema formation and subsequent regeneration. This is an exciting finding as the wound epidermis is thought to be essential in a variety of regenerative contexts, but is relatively unexplored and in general lacks molecular characterization (both in planarians and in other species). This is an important paper that does a nice job of identifying a gene via NGS and then functionally characterizing its role. These functional experiments are important as there are limited studies implicating a molecular player from the wound epidermis in any species. In general, the paper is well written and the data are presented and interpreted well. It could use some restructuring (i.e. a couple more figures and less extended data) and transitions throughout to help with readability. There are a few claims that need to be clarified. Further, in some places due to the complexity of the experiments it would benefit to have better diagrams/clearer text to convey the findings.

Major comments:

1. If making claims about cross phyla presence of equinox it's important to do more thorough orthology analysis than reciprocal BLAST. Ideally looking at synteny (which is not always possible given a lack of a quality genome assembly for many species) or a phylogeny-based orthology prediction (for a quick look shoot.bio) would be informative. Given this is the first description of the gene it's important that the information provided for potential orthologs in other species is as accurate as possible.
2. Single-cell RNAseq data: Lane 2 seems to be quite a bit lower than the rest of the lanes in regards to nUMI and nGene/ cell so calling those "similar" is not accurate. Why is lane 2 so low? It seems like the mean reads per cell is actually quite high so why are there so few genes and UMIs? Are a lot of these reads going unmapped? Further, while viable cells were sorted for scRNAseq it would be good to do some further QC, such as percent mitochondrial reads, to attempt to identify any further damaged/dying cells in the dataset. In addition, please clarify on how you sequenced the 10x libraries, notably explain the "28 x 40 paired end reads". Does this mean the biological read was 40 bp? Typically 10x libraries are sequenced with a ~90bp biological read so a variation on that requires explanation as shorter reads could be leading to reduced alignment rates. It's also not clear why lane 5 was included in the experiments (the RNAi control) and why intermediate calcein staining was taken for this lane.
3. Single-cell RNAseq data: It's not completely clear the general proportion of cell types at the different time points. For example, are all clusters in Figure 2D composed of cells from all time points/lanes? And if yes, was there any batch correction used? Further, it's noted that equinox is expressed in all epidermal cells throughout regeneration, but this is difficult to see from the single-cell RNA seq data presented. For example, in Figure 2E there are very few equinox+ cells in the mature epidermis and you can't tell if they come from 0, 6, or 18hpa. The in situ performed do a good job of confirming presence of equinox in these populations but the single-cell data could be presented in a manner that breaks down equinox expression. Further, there are different time points sampled, but there isn't much use of these data over time. It would be nice to leverage this dataset to understand more of how equinox expression is changing/turning on in these individual cell types over time or differentiation (either of known cell types over their order of differentiation or pseudotemporal ordering and tracking gene expression).

Minor comments:

1. In the abstract it is stated that equinox is a secreted protein and while it has a predicted signal sequence and is likely secreted there are no data confirming that it is secreted.
2. In the introduction it's a bit confusing whether neoblasts also regenerate the epidermis. I believe this is addressed later on, but it would be good to clarify it here as well.
3. The experiment setup in Figure 1 is confusing. Figure 1B is RNAseq on anterior facing wounds from tails and then authors note that these findings were validated by Figure 1C which are different types of wounds (though anterior and posterior-facing). It should be clarified how these confirm the RNAseq data as the figure legend and text don't make it clear why both injuries are being shown and which is confirming the RNAseq data. One would expect a like-for-like validation with the markers from 1C used on tissue like in 1D, E, and G.
4. For supplementary tables with gene names, these should not be in excel as gene names are converted to dates.
5. It would be nice to give some rationale about why higher doses of bmp4 RNAi were explored and potentially some intro on bmp signaling. Also, does RNAi target certain areas of the animal more effectively? Or do you expect that only a small amount of expression of bmp4 is required for blastema formation vs. ML regeneration? Further, in the previously published low dose bmp4 RNAi is equinox expression affected?
6. Wnt1 and follistatin both actually seem to be up earlier in the bmp4 RNAi samples (Figure 1B). Can you comment on this and potential altered kinetics of the wound-induced genes?
7. Can you comment on the thickening of the wound epidermis in planarians. After 6 hours the wound epidermis appears to be multiple cell layers thick whereas in vertebrates it would still be only one cell layer thick. Is there massive migration happening? Looking at Figure 2g here which really emphasizes the thickness of the epidermis.
8. Bulk RNA seq analysis methods: what is the read length used? If over 50bp reads why use bowtie instead of bowtie2?
9. Line 194: Is there a citation for "smedwi-1 transcripts can be residually detected in post-mitotic progenitors" or is this coming from this paper and if so where?
10. Do equinox RNAi animals fail to make a wound epidermis? (e.g. if you repeat Fig 2C with RNAi does it show a similar phenotype to control?)
11. It's shown that neoblasts regenerate tissue like muscles and neurons in equinox RNAi animals even though these animals fail to form blastemas. An interesting extension (which should NOT be a required experiment for a revision given that the current claims are supported) would be if a single neoblast that is activated in equinox RNAi animals could be transplanted and fully recover an irradiated animal.
12. Are Figure 2D and extended data Figure 4C the same graph?
13. Line 287: equinox+ progenitors is used here, but this is a bit dangerous as you've previously shown it's expressed by quite a bit of cells and not solely progenitors.
14. It would be nice to elaborate a bit more about blastema-independent regeneration when equinox is knocked down.

Signed by: Nicholas Leigh

REVIEWER COMMENTS

Reviewer #1 (Remarks to the Author):

In the manuscript 'The planarian wound epidermis gene *equinox* promotes positional information resetting to initiate regeneration' Scimone, Cloutier et al. identify a new gene, *equinox*, which could be a target of *bmp4* in promoting the regenerative response. *equinox* RNAi planarians fail to reestablish positional information, to upregulate proliferation and do not regenerate a blastema. The authors propose that *equinox* could be an early signal in epidermal-muscle cell communication following wounding in planarians, similar to the role of the wound epidermis seen in vertebrates.

The finding and characterization of this new gene is relevant in the field of regeneration and biology in general. The work is of high scientific quality. However, the interpretation of some results is questionable. Before publication, the following concerns must be addressed.

We thank the reviewer for the comments and assessment of the work.

Major concerns:

1. The main concern of the presented study is the assumption that the inability to regenerate of *bmp4* and *equinox* RNAi animals results from the inability to reset positional information, which in fact is the title of the manuscript, and should be revised.

According to the results, *bmp4* and *equinox* RNAis do not only show problems in positional information but show a drastic decrease in the number of neoblasts near the wound during regeneration and a decrease in neoblast-associated genes. Although the authors argue that this is not the reason of the non-regenerating phenotype, an analysis of the dynamics of proliferation during the early and late regenerative phases is required, to test whether *bmp4* and *equinox* RNAis show a decreased or a delayed mitotic response. It could be that the inability to regenerate is linked to a problem in triggering the regenerative mitotic response.

We analyzed the mitotic response in *equinox* RNAi animals (new Fig. 5b) and similar to the results observed in the bulk sequencing data and the *smedwi-1* stainings for both *bmp4* and *equinox* RNAi animals shown in the paper, mitotic numbers were significantly reduced in *equinox* RNAi at 48 hpa but not at 6 hpa. We explained in the paper and further addressed it in the revised version (page 7, lines 198-202) that lack of neoblast proliferation cannot explain the absence of blastema. This is because it has previously been established in Tewari et al. 2018 that the lack of a neoblast proliferation peak at 48 hpa does not block regeneration, but only affects the time required to form a blastema (regeneration occurred without a proliferative 48 hpa peak, just more slowly). This is also the reason why we showed that there was still no blastema in either 20 dpa *bmp4* or 28 dpa *equinox* RNAi animals. At 20-28 dpa it would be expected that animals overcome a decreased or delayed mitotic response, as shown in Tewari et al 2018; by contrast, our results indicate that *equinox* and *bmp4* RNAi animals lack the ability to ever form a blastema. We show that neoblasts are present and capable of undergoing cell turnover in these fragments that show no regeneration (See Fig. 4b – 28 dpa: neoblasts are still present and undergoing cell turnover to replace tail cells but do not make head cells). We also added a more detailed description of the findings from Tewari et al. to the text to make this point clearer.

The arguments of the authors to claim that the proliferative response is not a cause of the non-regenerative phenotype are confusing. For instances, in Lines 253-257 it is read: 'The failure of regeneration in *equinox* RNAi animals was not a consequence of a lack of neoblast maintenance and differentiation capacity, because neoblasts of tail fragments that did not regenerate by 28 dpa were still able to differentiate into tail neurons and muscle cells (Fig. 3b), i.e., neoblasts were capable of making pre-existing tissues but did not regenerate those that were missing.' This argument is not valid, since it could be that *equinox* is not required for neoblast maintenance and differentiation, and that's the reason why some structures are differentiated several days after the cut, but it could be required for the mitotic response of neoblasts at initial regeneration stages. This is indeed a main claim of the manuscript, that *equinox* has a specific role during regeneration but not in the maintenance of tissues.

As stated above, the lack of proliferation only affects the rate of blastema formation, but not the ability to form a blastema (Tewari et. al, 2018). With the 28 dpa experiment, we showed that the problem in *equinox* RNAi animals is not a consequence of a generic neoblast defect, because neoblasts in these animals were

still able to differentiate into several tissues to homeostatically replace tissues, but are unable to form missing tissues in the blastema outgrowth.

Even more, the observation that in *bmp4* and *equinox* RNAi the brain, eyes, pharynx... can be regenerated in the pre-existent tissues suggests that the problem is not the positional information but the inability to create enough new cells to make a blastema where to differentiate the missing structures.

equinox RNAi animals did not regenerate anterior tissues in the pre-existent tissues, only *bmp4* RNAi animals did this. This is most likely because some small amount of *equinox* was still expressed under *bmp4* RNAi conditions and some positional information re-scaling still occurred in some animals. However, in both RNAi conditions at posterior-facing wounds we observed some posterior tissues such as the posterior pole. Regardless, as discussed above, a blastema can be made without a second mitotic peak.

In the same direction, the observation that *bcat1* RNAi does not rescue the phenotype further suggests that the problem is not positional information but it could be proliferation. The same occurs with the results showing that small but not big injuries can be regenerated. For instances, an eye can be regenerated, but not a pharynx, which demands much more many cells, although neither of them is different in the requirement of a reset in the positional information.

We now performed FISH experiments in double *b-catenin*; *equinox* RNAi animals and found that *b-catenin* RNAi could indeed cause *equinox* RNAi animals to express anterior positional information, but nonetheless still failed to form a blastema (new Fig. 6c). These results indicate that resetting of positional information in *equinox* RNAi animals is not sufficient to restore blastema formation. Without a blastema, we found evidence for some differentiation of anterior cell types in preexisting tissues in *b-catenin*; *equinox* RNAi animals. To explore these observations further, we also performed new experiments using the Erk signaling inhibitor PD0325901 (Owlarn, 2017). This Erk signaling inhibitor blocks wound signaling, rescaling of positional information, upregulation of neoblast proliferation, and formation of a blastema. We used this inhibitor in *b-catenin* RNAi animals and found that these animals were also able to rescale positional information and differentiate new tissues in the pre-existent tissues in the absence of blastema formation, further indicating that positional information re-setting is insufficient to restore blastema outgrowth in Erk inhibitor-treated animals (new Suppl. Fig. 13). We added these figures to the manuscript, and changed the title to reflect these intriguing findings. Prior work on *folliculin* RNAi animals (Gaviño et al. 2013, Roberts-Galbraith et al. 2013, Tewari, 2018) and *myoD* RNAi animals (Scimone et al. 2017) has shown that re-setting positional information can be required for regeneration. However, in these cases addition of *b-catenin* RNAi restores regeneration. Our findings indicate two defects in *equinox* RNAi animals: positional information re-setting and blastema outgrowth. Our prior work in the original submission also supported this conclusion (specifically, failed blastema formation despite PCG expression regeneration in early RNAi timepoints for *bmp4* and failure of *b-catenin* RNAi to suppress blastema formation in *equinox* RNAi). For example, " These results, with short RNAi conditions, suggest that *bmp4*, possibly through *equinox*, is required for blastema growth itself, in addition to the MTR and PCG expression regeneration." We feel these new results substantially support this prior conclusion and therefore add important new data to the paper. We thank the reviewer for the question, which spurred us to investigate the *b-catenin*; *equinox* double RNAi conditions at the molecular and cellular level.

We also further analyzed pharynx regeneration. Previously we have assessed the pharynx regeneration but adding chloretone and looking at pharynx protrusions. Now we analyzed with markers by FISH and found that the pharynx regeneration occurred and updated the paper accordingly.

2. The second main concern refers to whether the early wound response is affected in *bmp4* or *equinox* RNAi. The authors claim that early wound response takes place in *bmp4* or *equinox* RNAi. However, according to the RNA-seq data and to the in situs, the initial wound response in *bmp4* and *equinox* RNAi animals is not a normal response. In general, a decrease of the wound-induced genes is seen in the RNA-seq (fig 1b and 3c). In fact, *equinox* is chosen because it is downregulated during the first hours after *bmp4* RNAi, as also seems to occur with *inhibin*, *glypican* or *fos-1*. And, clearly, the pattern of expression of the wound genes at 6h is not the same in *bmp4* or *equinox* RNAi

animals than in controls. Thus, according to the results showed, the early wound response is already affected in these phenotypes.

We now added a new heatmap showing expression of early wound-induced genes for both *bmp4* (Suppl. Fig. 2a) and *equinox* RNAi (Fig. 4c) at different timepoints following amputation. Expression of early wound-induced genes is activated following inhibition of *bmp4* and *equinox* (Suppl. Table 1). Moreover, we tested expression of some of them by FISH in *equinox* RNAi animals at 3 hpa and even though there might be a reduction in the expression of *fos-1* and *egr-2* those genes are still expressed.

Similarly, we now added a heatmap showing expression of muscle-wound induced genes that generally peak in expression at around 16 hpa. Expression of some of these genes are already significantly downregulated at 6 hpa in both *bmp4* and *equinox* RNAi animals, and most of them are significantly reduced at 16 hpa. FISH expression experiments validated these trends at both 6 hpa and 16 hpa under both RNAi conditions, showing decreased expression of muscle wound-induced genes. We updated the text to indicate that expression of some of these muscle wound-induced genes was lower in *equinox* RNAi animals by 6 hpa.

Furthermore, along the manuscript it is not clear whether the authors consider that *bmp4* and *equinox* exert a role during the first hours of regeneration or not. For instances, when comparing with *follistatin* or *myoD* RNAis they also conclude that 'the regeneration-specific defect in *bmp4* RNAi animals might therefore involve a previously uncharacterized process associated with regeneration initiation.' This aspect of the manuscript is very confusing and should be clarified. (An advice would be also to omit the comparison with the *follistatin* or *myoD* RNAis, since it just increases complexity but not provides any answer to *equinox* function.)

We appreciate the idea of this comment and understand the perspective. However, in the end we feel it is important to compare these data with *follistatin* and *myoD* RNAi conditions because these three genes are all required for regeneration but not tissue turnover. Therefore, comparing the underpinnings of the phenotypes helps understanding the requirements for regeneration initiation and the explanation for the regeneration failure phenotypes of *bmp4* and *equinox* RNAi animals.

For instance, *myoD* RNAi animals have reduced numbers of longitudinal fibers and therefore, reduced wound-induced expression of wound-induced *notum* and *follistatin*, as well as decreased perdurance of expression of other muscle wound-induced genes. *follistatin* RNAi animals, by contrast, display a defective missing tissue response; ie. no neoblast peak in proliferation at 48 hpa, no elevated apoptosis at 72 hpa, however in this case, RNAi animals can still regenerate blastemas at a slower rate. Amputation of *follistatin* RNAi animals at certain planes leads to failed head regeneration. However, *follistatin* RNAi (unlike *equinox* RNAi) causes upregulation of wound-induced *wnt1*; it is this aspect of the *follistatin* RNAi phenotype that causes head regeneration failure. Specifically, *wnt1* RNAi suppresses the head regeneration defect of *follistatin* RNAi phenotype - though regeneration still occurs without the second proliferative peak. This latter point helps for instance in illustrating that the second mitotic peak is not essential for blastema formation and that this aspect of the defect in *equinox* RNAi animals cannot simply explain regeneration failure (addressing point 1 above). *equinox* RNAi animals have a broader impact on wound-induced gene expression than do these other genes and fail to rescale positional information, and do not form a blastema even at later time points.

3. In planarians the early wound response includes the expression of the wound-induced genes, the activation of mitosis and also the activation of apoptosis. In the previous point 1 the importance of analyzing mitosis has already been exposed. To clarify whether the early regenerative response takes place (previous point 2), the apoptotic response should be also analyzed. It is relevant to understand whether the non-regenerative phenotype described is associated to a failure in triggering the very early apoptotic response, directly associated to the injury, or to the late apoptotic response, associated to the integration of the new and the old tissue, which could be linked to the appearance of differentiated structures in the pre-existent tissue showed in the RNAis.

We analyzed the apoptosis response in *equinox* RNAi animals (new Suppl. Fig.12a) and we did not observe any defect in apoptosis activation.

4. May be due to the compact form of the manuscript, there are some concepts that are not explained clear enough for common readers.

Line 44, an explanation of what is a blastema is required.

We now explain what a blastema is in page 2 Line 44.

“a blastema – an outgrowth that forms at the wound site and where differentiation of many missing tissues takes place.”

Lines 47-49 ‘Once the wound is covered, epithelial cells proliferate to form a thick epithelium, mobilization of progenitors occurs (involving tissue resident stem cells and/or dedifferentiation), and the regenerative blastema forms’. It is confusing if it refers to vertebrates, planarians... Lines 80-87, the introduction of PCGs is not clear enough for non-planariologists.

The statement refers to vertebrates. We modified the text to clarify this (Lines 48-55).

Lines 105-107, the concept of DV boundary is not properly explained.

We now added an explanation of the DV boundary concept (page 4 Lines 101-103).

“The planarian body plan includes specialized cells and patterns of gene expression at the dorsal-ventral median plane at the animal margin (the lateral edge), sometimes referred to as the DV boundary (DVB).”

5. Since equinox is a new gene, a better phylogenetic analysis is needed, including more species and showing the alignment and the tree. Some questions can be answered with it. For instances, do some species that have Thrombospondins also have Equinox, or they only have one of them at a time? Is it a correlation between having echinox and the regeneration capacity?

Regarding Thrombospondins, have them been related with BMP signal in any report?

Phylogenetic analysis of extracellular matrix proteins that contain different domain architecture can be challenging; phylogenetic studies often need to be constrained to particular domains, and often particular domains are lacking or additional domains are present and this information is not well assessed. In such instances, comparison of domain architecture is an alternative approach for assessing evolutionary relationships. Because the EGF domains in Equinox show similarity by blast to EGF domains in genes named "VWDE", we decided to now include this protein in the domain structure analysis (new Fig. 2a). Moreover, we also ran phylogenetic analyses using the EGF domains only and found that Equinox and VWDE proteins resolved into different groups (new Suppl. Fig. 4). We updated the text to include more discussion of the evolution of this protein family. There are some studies showing a connection of thrombospondins and bmp signaling but because Equinox lacks other domains of the Thrombospondin family we did not pursue this further.

6. Are *bmp4* and *equinox* co-expressed in the DV boundary? And in the blastemas? Is *bmp4* expressed in *equinox* RNAi animals? There are very direct questions that should be answered in the manuscript.

***bmp4* is well established to be expressed in planarian muscle cells, whereas *equinox* is largely expressed in epidermal cells. Therefore, they should not be co-expressed. Furthermore, *bmp4* is expressed most strongly dorsal-medially - its pattern is graded to low laterally. Nonetheless, we performed double *in situ* hybridizations for *equinox* and *bmp4*, but because of technical challenges at present with available anti-FITC antibodies we were unable to obtain publication quality data. Instead, we performed analysis on single-cell RNA sequencing data: the Pearson correlation R score was -0.01 indicating there is almost no overlap in expression in our single cell data between *bmp4* and *equinox*. These graphs are shown in Suppl. Fig. 5f. *bmp4* was unaffected in *equinox* RNAi animals (Suppl. Table 3)**

7. *equinox* seems to be a target of *bmp4*, and it shows a very specific dorsal localization. However, *equinox* RNAi animals do not show DV defects. Is *equinox* a *bmp4* mediator only in the regenerative response? And this response is not linked to the role of *bmp4* in polarity? A further discussion is required.

We don't think *equinox* mediates *bmp4* action in dorsal-ventral patterning of tissues generally because, as the reviewer noticed, there were no detectable DV defects in *equinox* RNAi animals (Suppl. Fig. 10e). We know that *equinox* expression is enriched in epidermis just dorsal to the DV median plane, and therefore, because *bmp4* inhibition causes ventralization (the replacement of dorsal fates with ventral cell identity), we hypothesize that *equinox* expression is lost during cell turnover and during regeneration because of loss of dorsal epidermal fate in *bmp4* RNAi conditions. However, we cannot rule out a direct role of *bmp4* on *equinox* expression. We now added more discussion about this in lines 295-306.

Minor concerns:

1. Scheme in figure 4- What are the pale pink cells in the wound that seem not to express *equinox*? According to figure 2c, *equinox* is expressed in several lines of cells in the wound, not only in the ones contacting the pre-existent tissue. It is confusing.

As labeled in the key of the model, all pink cells represent wound epidermis that express *equinox*. In Figure 3, we show that *equinox* is expressed in wound epidermis and in epidermal progenitors, which means there is not a single cell layer of *equinox* expression, but for a simple schematic we only labeled the epidermis layer.

2. How can be explained the asymmetry of the poles when they can regenerate? Some comment about that would be helpful.

As has been previously shown in Oderberg et. al, 2017, poles required cues from existing tissues to be formed. One of those clues is the DV boundary. In the absence of new DV boundary in the regenerating *bmp4* or *equinox* RNAi fragments, occasionally the pole forms at the old DV boundary, and therefore it is not at the midline. We agree that this is an interesting feature and note that in the text (lines 119-130).

3. Figure 3b. Control is missing. It is necessary for readers.

We now added it in Suppl. Fig. 10b.

4. Lines 118-120. 'Anterior poles formed more often at *bmp4* RNAi anterior-facing wounds in the animal anterior than in the posterior (e.g., trunk fragments, Extended Data Fig. 2c), but these wounds still lacked blastema outgrowth. Posterior poles formed frequently, even though no blastema was observed at *bmp4* RNAi posterior-facing wounds (Extended Data Fig. 2a, c)'. It is not clear what means posterior, what is more frequent... It should be clearly explained. Specifying that it is based in the expression of *notum* in A and *wnt1* in P will help.

We are comparing the frequency of animals displaying anterior pole formation in trunk fragments (with a transverse wound in the anterior) versus tail fragments (with a transverse wound in the posterior). Whereas in trunk fragments the anterior poles frequently formed in *bmp4* RNAi animals, coalescence of anterior poles in tail fragments was less frequent (a smaller number of tail fragments were able to form an anterior pole in *bmp4* RNAi animals). We rephrased this explanation in lines 112-119.

5. It is claimed that the DV boundary can be recognized by the expression of *laminb*. However, according to image 2b, *laminb* seems to be ventral to the DV boundary. Furthermore, it is very interesting that the expression of *laminb* and *equinox* seem to be separated by 1 cell. Is it always like that?

We generally refer to the DV boundary to the region where the dorsal and ventral epidermis meet (lines 101-103). Historically, *lamin B* has been used as a marker for this location. We now find a gene, *equinox*, that is expressed in that region, however dorsally to *lamin B*. There is not always a single cell separation between cells positive for these markers (*equinox* and *lamin B*).

6. The expression of *echinox* in intact animals is clearly dorsal. Has it also a dorsal pattern in the regenerating wounds? In images in 2c it seems that it is both, in D and V, although the in situ and the SC-seq data shows that at 18h it is more dorsal than ventral. At some point during regeneration it must be relocated in the dorsal part. When does it happen? It could be that during the first hours *equinox* is delocalized and then some upstream signal restricts it to dorsal. This could be linked to the different role that it exerts while it is delocalized with respect to its localized expression in intact animals.

Expression of *equinox* is biased to the dorsal side. It is technically hard to determine the origin (dorsal versus ventral) of the epidermal cells expressing *equinox* at 6 hpa. The pattern of dorsal and ventral identity and how this resolves over time in blastema formation is poorly understood. This could be an interesting target for future work. The sc-RNAseq data during regeneration also shows a bias in expression to dorsal progenitors.

Reviewer #2 (Remarks to the Author):

In the manuscript, the authors present equinox, a gene encoding a likely secreted protein with several conserved domains, and a strong requirement for regeneration initiation in planarians. They identified equinox in a search for BMP4-dependent genes after they showed that BMP4 RNAi had more severe and general effects on planarian regeneration than previously published.

The authors convincingly show that equinox RNAi specifically blocks regeneration at an early stage, while it does not affect homeostatic turnover in non-injured planarians. They characterized its expression pattern and show it is activated in various cell types, predominantly in epidermis and epidermal progenitor cells. Based on the expression pattern, they hypothesize equinox may be involved in epidermis-muscle communication as part of a wound epidermis signaling program initiating regeneration.

This gene is truly interesting as it is relatively uncharacterized and has a very strong regeneration phenotype that is specific for regeneration initiation.

We thank the reviewer for the comments and assessment of the work.

However, there are two major points I have to raise:

1) The authors state that equinox is not present in vertebrates. I did a quick Blast search with the equinox sequence, and found Van Willebrand Factor D and EGF Domains (VWDE), conserved in vertebrates. VWDE was recently identified by the Whited lab as a secreted protein induced in different cells in regeneration blastemas of several regeneration competent animals and required for limb regeneration in axolotl (<https://pubmed.ncbi.nlm.nih.gov/32163674/>).

As equinox and VWDE are best Blast hits in reciprocal Blast searches, they have a similar function in regeneration, and have a regeneration-associated expression pattern, it is very likely that equinox is a functional homolog of VWDE. It is important that the authors clarify the evolutionary relationship with VWDE and cite the study.

We agree that the similarity to VWDE is interesting to assess, and we therefore now added a more comprehensive analysis of Equinox and VWDE in Fig. 2a and Suppl. Fig. 4. Naturally, we noted the blast hits and the interesting role of VWDE in axolotl is intriguing. However, we noticed that the planarian protein lacked the von Willebrand domain that is a defining part of this gene name and this led us to identify another clade of proteins across animals that displays the same domain architecture of Equinox and also comes out in Blast searches. Equinox and these other proteins share thrombospondin repeats that are lacking in VWDE proteins.

Phylogenetic analysis of extracellular matrix proteins that contain different domain architecture can be challenging; phylogenetic studies often need to be constrained to particular domains, and often particular domains are lacking or additional domains are present and this information is not easily assessed in tree building. In such instances, comparison of domain architecture is an alternative approach for assessing evolutionary relationships. We decided to now include the VWDE protein in the domain structure analysis (new Fig. 2a), and we also added description of this in the text and reference the VWDE paper. Moreover, we also ran phylogenetic analyses using the EGF domains only and found that Equinox and VWDE proteins resolved into different groups, although the tree resolves with only moderate statistical support (new Suppl. Fig. 4). It is possible that the VWDE and Equinox families of proteins share an evolutionary history and/or there could prove to be similarity in function between these two clades of proteins, which will be an interesting target for future study.

2) The study seems rather preliminary. Are there any transmembrane proteins in stem cells/muscle cells/epidermal cells or blastema cells that could act as receptors for Equinox? The scRNAseq dataset might reveal some of these signaling components. Given that equinox/vwde's function in regeneration has been shown for other regenerating animals, identifying the receptor and downstream signaling pathways would add novelty to the study, especially, when it can be shown that these receptors or pathways are conserved in other regenerating animals.

We do have a variety of projects ongoing in the lab looking at the functions of transmembrane proteins in neoblasts and muscle cells. However, transmembrane proteins are involved in a myriad of processes from fate specification to stem cell renewal, to migration, to pattern of gene expression in muscle, to ECM interactions, etc, and substantial work is needed to parse the roles of transmembrane proteins into these various functions. Whether *Equinox* actually has a cognate receptor in one of these cell types would also only be a hypothesis at this point. It could have a different kind of role, such as regulating the diffusion of some other factor in the extracellular matrix. Therefore, we conclude that searching for a signaling cascade that could be required for *equinox* function, although very interesting, escapes the scope of this study.

Other points:

3) I find the BMP4 phenotype not convincing enough to speak of a general early role of *bmp4* in planarian regeneration. In Fig. 1, at posterior blastemas, there is not much difference in gene expression in control and *bmp4* RNAi animals. Is there a way to quantify the reduced expression of regeneration genes, or do the authors have RNAseq data on posterior blastemas similar to what they show for anterior blastemas in Fig.1b?

We showed now *bmp4* RNAi animals do not regenerate a posterior blastema at day 20 post amputation (Fig. 1a), and defects in forming a tail can also be observed in Suppl. Fig. 1b. We modified the text to indicate that despite this blastema formation defect there was posterior pole formation in pre-existing tissues. Unfortunately, we do not have publication quality RNAseq data for posterior blastemas.

In Ext. Data 2e, it looks like there is quite a big blastema (unpigmented tissue) that even contains eye spots? The posterior-facing wound is more convincing here.

Incomplete *bmp4* inhibition (weak RNAi, or 1 feeding RNAi as shown in Fig. 6a) allows animals to express some *equinox* as well as other wound-induced genes (Suppl. Fig. 13a, b), allowing for re-scaling of positional information and regeneration of missing tissues in pre-existing tissues (Fig. 6a, b). Sometimes, small blastemas can form. Under stronger *bmp4* inhibition conditions (more time post-initiation of RNAi), no blastemas can form at all, and no regeneration of anterior tissues in pre-existing tissues frequently occurred.

In the now Suppl. Fig. 3a, we amputated in the very anterior and still see defects in blastema formation but more anterior tissues are formed. This makes sense because less anterior PCGs are missing and anterior progenitors are present in this location.

4) The authors tested a requirement for BMP4, follistatin and MyoD for *equinox* expression. Only *bmp4* RNAi had an effect. Since there is a very early and conserved requirement for ERK signaling in regeneration, yet the downstream effectors are not known, it would be interesting to see if *equinox* activation depends on this pathway.

We now performed this experiment (new Suppl. Fig. 7c) and observed that *equinox* expression was inhibited in animals treated with an Erk signaling inhibitor (PD0325901).

5) To my unskilled eye, the H&E staining in Fig. 2 shows an open wound and does not seem to be covered by any epidermis. Staining with an epidermal marker would be helpful, or at least a close-up of the H&E images, so that epidermal cells can be identified by morphology.

We now added new panels (new Fig. 2c) using epidermal-specific markers (*rootletin* and *PRSS12*) and with this approach the epidermis was observed almost completely covering the wound at 18 hpa.

6) Fig. 3c: *equinox* expression is induced during the first 3h after injury in *equinox* RNAi planarians. Do the authors have an explanation for this?

The *equinox* expression observed at 3 hpa in the *equinox* RNAi heatmaps is a result of *equinox* dsRNA in the system being detected in the RNA sequencing. This has been seen in previous studies (Scimone, 2017). This

is variable and probably correlates with how much dsRNA from feedings is still in the intestines.

7) Fig. 1b: the time scale is missing on the x-axis.

Thank you for noting this. It has now been added.

8) No percentage of cells that express the gene in the different Seurat groups in fig. 2e, 2h.

We now added dot plots showing percentages in Fig. 3a, 3e, and Suppl. Fig. 5f.

9) On some occasions, FISH is used in the text to refer to what looks like non-fluorescent ISH to me (fig1c, 1d, 1e3d, 3e, 3f, 4b)

We performed FISH in all instances, but the panels referred here are pseudo-colored in black (black and white images) to facilitate visualization.

10) SMEDWI1 (Fig. 2g) is not mentioned in the main text.

We now mention SMEDWI-1 in line 281.

Reviewer #3 (Remarks to the Author):

This paper investigates the consequences of high dose bmp4 knockdown and through a series of bioinformatic and wet lab experiments identifies a role for a novel wound epidermis gene, equinox, in blastema formation and subsequent regeneration. This is an exciting finding as the wound epidermis is thought to be essential in a variety of regenerative contexts, but is relatively unexplored and in general lacks molecular characterization (both in planarians and in other species). This is an important paper that does a nice job of identifying a gene via NGS and then functionally characterizing its role. These functional experiments are important as there are limited studies implicating a molecular player from the wound epidermis in any species. In general, the paper is well written and the data are presented and interpreted well. It could use some restructuring (i.e. a couple more figures and less extended data) and transitions throughout to help with readability. There are a few claims that need to be clarified. Further, in some places due to the complexity of the experiments it would benefit to have better diagrams/clearer text to convey the findings.

We thank the reviewer for the comments and suggestions for the paper.

Major comments:

1. If making claims about cross phyla presence of equinox it's important to do more thorough orthology analysis than reciprocal BLAST. Ideally looking at synteny (which is not always possible given a lack of a quality genome assembly for many species) or a phylogeny-based orthology prediction (for a quick look shoot.bio) would be informative. Given this is the first description of the gene it's important that the information provided for potential orthologs in other species is as accurate as possible.

We agree that more presentation on the orthology analysis would be appropriate. By Blast we had noted similarity between Equinox and two families of proteins. However, we found that Equinox shared domain structure best with one of these families. The other family, VWDE, is of interest because a role for this protein in *Axolotl* and *Xenopus* regeneration has been shown. However, we noticed that the planarian protein lacked the von Willebrand domain that is a defining part of this gene name and this led us to identify another clade of proteins across animals that displays the same domain architecture of Equinox and also comes out in Blast searches. Equinox and these other proteins share thrombospondin repeats that are lacking in VWDE proteins.

Phylogenetic analysis of extracellular matrix proteins that contain different domain architecture can be challenging; phylogenetic studies often need to be constrained to particular domains, and often particular domains are lacking or additional domains are present and this information is not easily assessed in tree building. In such instances, comparison of domain architecture is an alternative approach for assessing evolutionary relationships. We decided to now include the VWDE protein in the domain structure analysis (new Fig. 2a), and we also added description of this in the text and reference the VWDE paper. Moreover, we also ran phylogenetic analyses using the EGF domains only and found that Equinox and VWDE proteins resolved into different groups, although the tree resolves with only moderate statistical support (new Suppl Fig. 4). Synteny wasn't an attractive approach here given the distance of the planarian genome from some of the other clades assessed. It is possible that the VWDE and Equinox families of proteins share an evolutionary history and/or there could prove to be similarity in function between these two clades of proteins, which will be an interesting target for future study.

2. Single-cell RNAseq data: Lane 2 seems to be quite a bit lower than the rest of the lanes in regards to nUMI and nGene/ cell so calling those "similar" is not accurate. Why is lane 2 so low? It seems like the mean reads per cell is actually quite high so why are there so few genes and UMIs? Are a lot of these reads going unmapped? Further, while viable cells were sorted for scRNAseq it would be good to do some further QC, such as percent mitochondrial reads, to attempt to identify any further damaged/dying cells in the dataset.

We agree with the reviewer that lane 2 is of lower quality than the other lanes as seen in the QC presented in Suppl. Fig. 5a, b, and c. We removed the word 'similar' when describing the contribution of the lanes to the UMAP plot in Suppl. Fig. 5.

Lane 2 represents cells collected from wounds at 6 hpa. We decided to collect both calcein high and calcein intermediate cells for this time point, all of which were PI negative. We wanted to capture as many wounded cells as possible as we suspected *equinox* was expressed in wounded cells at this time point. As wounded cells would be more likely to have compromised membranes at early time points, we reasoned that they might have calcein intermediate levels. It is possible that some of these cells then underwent apoptosis between isolation with FACS and the scRNA-seq experiment, resulting in a lower quality lane. A lot of these reads went unmapped as suggested by the reviewer. Reads mapped confidently to genome per 10x were 83.2%, 57.7%, 83.8%, 80.5%, and 81.8% for lanes 1-5 respectively. When inspecting the unmapped reads some of these unmapped reads were mitochondrial in nature. Because our current transcriptome only includes some but not all mitochondrial genes (several mitochondrial rRNA genes are not in the v6 version of the transcriptome) some reads to these lacking mitochondria genes will go unmapped.

We were able to assess mitochondrial read burden using 10/15 mitochondrial genes that are included in the transcriptome. There were higher mitochondrial reads in lane 2 as suggested by the reviewer for reasons detailed above. We have now included this analysis in the manuscript (new Suppl. Fig. 5c).

We decided to include this lane, however, because the cells still expressed expected genes without inappropriate overlap of genes known to be mutually exclusive between cell types. We also noted that these cells were still able to cluster both based on cell type and differentiation state, so could be useful in further identifying *equinox* expression. We also validated all observations made with scRNA-seq related to *equinox* expression by in situ hybridization, which added confidence to the reported results. We decided to explicitly point out that this lane was of lower quality in the figure legends and methods, and to explain our rationale for using this data, involving key observations on expression being validated by FISH.

In addition, please clarify on how you sequenced the 10x libraries, notably explain the "28 x 40 paired end reads". Does this mean the biological read was 40 bp? Typically 10x libraries are sequenced with a ~90bp biological read so a variation on that requires explanation as shorter reads could be leading to reduced alignment rates. It's also not clear why lane 5 was included in the experiments (the RNAi control) and why intermediate calcein staining was taken for this lane.

The Whitehead sequencing core suggested usage of 28x40 bp over 28x90 bp reads, after they completed an internal process of optimizing 10X sequencing experiments for their core that involved sequencing on a HiSeq. The specifications on the 10X website were optimized on the NextSeq which has different kit configurations, and in their conversations with the 10X company this read length was deemed reasonable for the approach being used. It is now standard in our core.

Lane 5 (which was fed control RNAi) was included in the experiment so that our dataset would have more cells, and therefore more power – not for an explicit experimental reason involving control dsRNA. We reasoned that food containing control dsRNA encoded by the *C. elegans* gene *unc-22* would be similar enough to regular food to not make a difference in cell expression which would change our overall results. These results were then further validated by FISH experiments. We now more explicitly describe that this lane was included to simply increase cell number in the text (methods).

Intermediate calcein was taken only for lane 2, which was originally listed as lane 5 in the methods. The rationale for this is described above.

3. Single-cell RNAseq data: It's not completely clear the general proportion of cell types at the different time points. For example, are all clusters in Figure 2D composed of cells from all time points/lanes? And if yes, was there any batch correction used? Further, it's noted that *equinox* is expressed in all epidermal cells throughout regeneration, but this is difficult to see from the single-cell RNA seq data presented. For example, in Figure 2E there are very few *equinox*+ cells in the mature epidermis and you can't tell if they come from 0, 6, or 18hpa. The in situ performed did a good job of confirming presence of *equinox* in these populations but the single-cell data could be presented in a manner that breaks down *equinox* expression.

We now added plots that split the data (the cells contributing to each cluster) by time point. This was done for the analyses that involved all cell types (new Suppl. Fig. 5d), and the analyses that were restricted to the epidermal lineage (new Suppl. Fig. 6b), including for *equinox* expression (new Suppl. Fig. 6c)

Seurat 3 is known to be one of the better pipelines for reducing batch effects as it uses CCA dimensionality reduction (<https://genomebiology.biomedcentral.com/articles/10.1186/s13059-019-1850-9>). Given that each major cluster contained cells from each lane (new Suppl. Fig. 5d), and we did not want to remove true biological variability as the lanes were acquired at 3 different time points. Therefore, we did not do any additional batch corrections. Our new Suppl. Fig. 6c showed *equinox* expression split by time point and maturation state, however, we agree with the reviewer that our FISH experiments are key for validating these results.

Further, there are different time points sampled, but there isn't much use of these data over time. It would be nice to leverage this dataset to understand more of how *equinox* expression is changing/turning on in these individual cell types over time or differentiation (either of known cell types over their order of differentiation or pseudotemporal ordering and tracking gene expression).

The epidermal lineage has previously been studied using scRNAseq data in Wurtzel et al 2017, which is why we felt more confident in naming epidermal lineage stages – and then correlating *equinox* expression with those stages to determine when in differentiation *equinox* is expressed.

We added panels showing *equinox* expression where Seurat plots are split by time point as described above (new Suppl. Fig. 6c), which will now allow the reader to correlate both time point (number of hours post amputation) and differentiation state as described in Wurtzel et al 2017.

We attempted to plot our data in Seurat using Monocle in order to further explore pseudotime applications with the data. If we used lanes from multiple time points, we found that Monocle's trajectories simply drew a line based on timepoint (ie 0 hpa -> 6 hpa -> 16 hpa), and not differentiation state. Therefore, we decided to try to use 16 hpa only lanes in order to explore whether Monocle could further elucidate *equinox* expression during cell differentiation. We were able to obtain a plot that grossly recapitulated the known differentiation pathway (below, with trajectories plotted on Fig. 3e – Fig. 3e shows the maturation stage labels annotated by use of marker genes previously characterized for these stages). As this did not really reveal new information that would be useful to further understanding the expression of *equinox*, we decided to not include it in the manuscript.

Minor comments:

1. In the abstract it is stated that equinox is a secreted protein and while it has a predicted signal sequence and is likely secreted there are no data confirming that it is secreted.

We modified the abstract wording (Line 29).

2. In the introduction it's a bit confusing whether neoblasts also regenerate the epidermis. I believe this is addressed later on, but it would be good to clarify it here as well.

Neoblasts differentiate into every cell type in planarians. Thank you for the suggestion, we now clarified this in the text (Lines 59-61).

3. The experiment setup in Figure 1 is confusing. Figure 1B is RNAseq on anterior facing wounds from tails and then authors note that these findings were validated by Figure 1C which are different types of wounds (though anterior and posterior-facing). It should be clarified how these confirm the RNAseq data as the figure legend and text don't make it clear why both injuries are being shown and which is confirming the RNAseq data. One would expect a like-for-like validation with the markers from 1C used on tissue like in 1D, E, and G.

In general, trunk and tail fragments behave similarly in planarian regeneration. For bulk sequencing experiments, we generally used tail fragments because re-scaling of positional information in tails is easily observed. These tail fragments have to re-pattern PCG expression from a region that initially only expresses posterior PCGs. The wound-induced gene expression assessment by in situ hybridization using trunks (Fig. 1d, Suppl. Fig. 2b, Fig. 4d, and Suppl. Fig. 11) allowed us to assess data simultaneously at both anterior- and posterior-facing wounds. We now added several new panels validating wound-induced gene expression findings in tail fragments (Suppl. Fig. 2b, and Suppl. Fig. 11a,b). We also modified the wording of the text to be clearer about the comparisons we are making.

4. For supplementary tables with gene names, these should not be in excel as gene names are converted to dates.

Thank you for this suggestion. We think keeping the data in excel tables will allow for easy filtering and usage by readers; we ensured the names were not converted to dates (cells can be converted to text).

5. It would be nice to give some rationale about why higher doses of bmp4 RNAi were explored and potentially some intro on bmp signaling. Also, does RNAi target certain areas of the animal more effectively? Or do you expect that only a small amount of expression of bmp4 is required for blastema formation vs. ML regeneration? Further, in the previously published low dose bmp4 RNAi is equinox expression affected?

Over time the RNAi protocol has improved. Previous experiments inhibiting *bmp4* involved bacteria expressing dsRNA and mixing of food with low-gelling agarose. We later dispensed with these aspects of the protocol and stronger gene inhibition emerged. We reference the current RNAi protocol used in the methods. RNAi is equally effective in every area of the animal examined, but different functions of a gene might require different thresholds of inhibition. Most likely, weak inhibition of *bmp4* is sufficient to impede normal ML regeneration but not enough to affect AP regeneration. By contrast strong inhibition of *bmp4* will affect both types of regeneration. There was not sequencing data from previous studies of *bmp* in planarians to look for *equinox* expression.

6. Wnt1 and follistatin both actually seem to be up earlier in the bmp4 RNAi samples (Figure 1B). Can you comment on this and potential altered kinetics of the wound-induced genes?

***wnt1* and *follistatin* were not significantly increased at earlier time points in *bmp4* RNAi animals (see below, p value)**

	dd_Smed_v6 id	control (unc22 RNAi) vs bmp4 RNAi											
		00 hpa		03 hpa		06 hpa		16 hpa		24 hpa		48 hpa	
		0_padj	3_log2FC	3_padj	6_log2FC	6_padj	16_log2FC	16_padj	24_log2FC	24_padj	48_log2FC	48_padj	
fst	dd_Smed_v6_9584_0_1	-0.2070662	0.4865344	0.2668913	0.1851198	-0.1093233	0.8857828	-0.4458593	0.00195891	-0.5926617	0.00000697	-0.4229537	0.00198441
wnt1	dd_Smed_v6_28398_0_1	0.7895336	0.3418654	0.8636632	0.05331461	-0.1452609	1	-0.9062162	0.01124925	-1.336531	0.0000174	-1.8300111	2.6873E-07

7. Can you comment on the thickening of the wound epidermis in planarians. After 6 hours the wound epidermis appears to be multiple cell layers thick whereas in vertebrates it would still be only one cell layer thick. Is there massive migration happening? Looking at Figure 2g here which really emphasizes the thickness of the epidermis.

Fig. 3d is looking with a dorsal view at the blastema – we added a cartoon to the figure to make this view clearer. Note that *equinox* is expressed not only in the wound epidermis but also in some epidermal progenitor cells underlying the epidermis. In this image many of the cells are epidermal progenitors underlying the outer epidermal layer. We modified the figure legend text to make this figure panel description clearer.

We now also added panels showing with a mature epidermis marker the lining wound epidermis in a cross-section (new Fig. 2c). It is easier to observe in this panel that mature epidermis at the wound is only one layer of cells.

8. Bulk RNA seq analysis methods: what is the read length used? If over 50bp reads why use bowtie instead of bowtie2?

The reads are 50 bp. Therefore, we used bowtie 1 instead of bowtie 2, because the utility of bowtie 2 increases with read length >50 bp and we wanted to compare data to previously published datasets (*myoD* and *fst* RNAi bulk sequencing) that were analyzed with bowtie 1.

9. Line 194: Is there a citation for “smedwi-1 transcripts can be residually detected in post-mitotic progenitors” or is this coming from this paper and if so where?

We now added a citation.

10. Do *equinox* RNAi animals fail to make a wound epidermis? (e.g. if you repeat Fig 2C with RNAi does it show a similar phenotype to control?)

We now added a new panel in Suppl. Fig. 10c that shows that *equinox* RNAi animals are able to covered the wound with epidermis similarly to control animals.

11. It's shown that neoblasts regenerate tissue like muscles and neurons in *equinox* RNAi animals even though these animals fail to form blastemas. An interesting extension (which should NOT be a required experiment for a revision given that the current claims are supported) would be if a single neoblast that is activated in *equinox* RNAi animals could be transplanted and fully recover an irradiated animal.

Thank you for the suggestion. Because neoblasts do not require activation for tissue turnover, being constantly dividing, we did not pursue this avenue.

12. Are Figure 2D and extended data Figure 4C the same graph?

This was the same graph depicting a key for which region of the plots correspond to which cell type for ease of assessing data. However, we now left it only in the main figure.

13. Line 287: *equinox*+ progenitors is used here, but this is a bit dangerous as you've previously shown it's expressed by quite a bit of cells and not solely progenitors.

Thanks for the suggestion. We modified the language in the text.

14. It would be nice to elaborate a bit more about blastema-independent regeneration when *equinox* is knocked down.

Thank you for the suggestion. We agree that this is an important point and we added an entire section on this point in the text: *equinox is required for blastema outgrowth*.

Indeed, we performed more experiments to further understand patterning in the absence of blastema formation. We now performed FISH experiments in double *b-catenin*; *equinox* RNAi animals and found that *b-catenin* RNAi could indeed cause *equinox* RNAi animals to express anterior positional information, but nonetheless animals still failed to form a blastema (new Fig. 6c). These results indicate that resetting of positional information in *equinox* RNAi animals is not sufficient to restore blastema formation. Without a blastema, we found evidence for some differentiation of anterior cell types in preexisting tissues in *b-catenin*; *equinox* RNAi animals. To explore these observations further, we also performed new experiments using the Erk signaling inhibitor PD0325901 (Owlarn, 2017). This Erk signaling inhibitor blocks wound signaling, rescaling of positional information, upregulation of neoblast proliferation, and formation of a blastema. We used this inhibitor in *b-catenin* RNAi animals and found that these animals were also able to rescale positional information and differentiate new tissues in the pre-existent tissues in the absence of blastema formation, further indicating that positional information re-setting is insufficient to restore blastema outgrowth in Erk inhibitor-treated animals (new Suppl. Fig.13). We added these figures to the manuscript, and changed the title to reflect these intriguing findings. Prior work on *folliculin* RNAi animals (Gaviño et al. 2013, Roberts-Galbraith et al. 2013, Tewari, 2018) and *myoD* RNAi animals (Scimone et al. 2017) has shown that re-setting positional information can be required for regeneration. However, in these cases addition of *b-catenin* RNAi restores regeneration. We also further analyzed pharynx regeneration. Previously we have assessed the pharynx regeneration but adding chloretone and looking at pharynx protrusions. Now we analyzed with markers by FISH and found that the pharynx regeneration occurred and updated the paper accordingly. Our findings indicate two defects in *equinox* RNAi animals: positional information re-setting and blastema outgrowth. Our prior work in the original submission also supported this conclusion (specifically, failed blastema formation despite PCG expression regeneration in early RNAi timepoints for *bmp4* and failure of *b-catenin* RNAi to suppress blastema formation in *equinox* RNAi). For example, " These results, with short RNAi conditions, suggest that *bmp4*, possibly through *equinox*, is required for blastema growth itself, in addition to the MTR and PCG expression regeneration." We feel these new results substantially support this prior conclusion and therefore add important new data to the paper.

Signed by: Nicholas Leigh

REVIEWERS' COMMENTS

Reviewer #1 (Remarks to the Author):

The authors have provided an answer to all concerns raised and the manuscript has been reviewed accordingly, being ready for acceptance.

Reviewer #2 (Remarks to the Author):

The authors have addressed all my main concerns.

One last comment: Line 244/45 'It is possible that dd_20318 and VWDE protein classes share an evolutionary history and/or some aspects of function, but this will require further investigation to assess.'

It is quite clear that they share aspects of function (and evolutionary history), this sentence should be more precise.

Reviewer #3 (Remarks to the Author):

The authors have done a great job addressing my comments and I recommend this manuscript move forward to publication. Nice work!

A tiny point:

1. In the response to reviewers it is stated that CCA was used in Seurat, but the methods do not make this clear. More details on the Seurat analysis including the addition of CCA as part of the analysis is important.

REVIEWERS' COMMENTS

Reviewer #1 (Remarks to the Author):

The authors have provided an answer to all concerns raised and the manuscript has been reviewed accordingly, being ready for acceptance.

We thank the reviewer for their efforts with our manuscript.

Reviewer #2 (Remarks to the Author):

The authors have addressed all my main concerns.

One last comment: Line 244/45 'It is possible that dd_20318 and VWDE protein classes share an evolutionary history and/or some aspects of function, but this will require further investigation to assess.'

It is quite clear that they share aspects of function (and evolutionary history), this sentence should be more precise.

We thank the reviewer for their efforts with our manuscript. We edited the sentence to make it simpler and clearer.

Reviewer #3 (Remarks to the Author):

The authors have done a great job addressing my comments and I recommend this manuscript move forward to publication. Nice work!

A tiny point:

1. In the response to reviewers it is stated that CCA was used in Seurat, but the methods do not make this clear. More details on the Seurat analysis including the addition of CCA as part of the analysis is important.

We thank the reviewer for their efforts with our manuscript. We added this information to the Methods Section.